# EFFICIENT INFERENCE WITH LARGE REASONING MODELS

## ABSTRACT

Large reasoning models (LRMs) achieve state-of-the-art performance by generating long chains-of-thought, but often waste computation on redundant reasoning after the correct answer has already been reached. We introduce Early-Stopping for Token-Aware Reasoning (ESTAR) that detects and reduces such reasoning redundancy to improve efficiency without sacrificing accuracy. Our method combines (i) a trajectory-based classifier that identifies when reasoning can be safely stopped, (ii) supervised fine-tuning to teach LRMs to propose self-generated <stop> signals, and (iii) <stop>-aware reinforcement learning that truncates rollouts at self-generated stop points with compute-aware rewards. Experiments on four reasoning datasets show that ESTAR reduces reasoning length by about x3.7 (from 4799 to 1290) while preserving accuracy (74.9% vs. 74.2%), with strong cross-domain generalization. These results highlight early stopping as a simple yet powerful mechanism for improving reasoning efficiency in LRMs. [1]

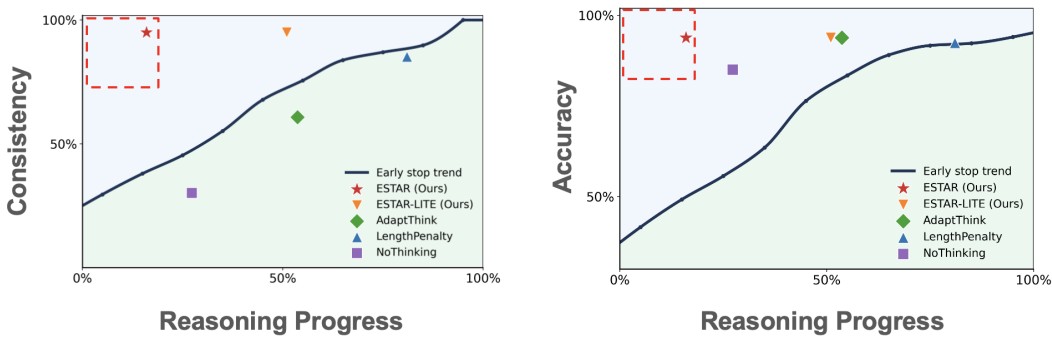

Figure 1: **Early-stopping chain-of-thought and its impact on the model's response**. The $x$-axis shows reasoning progress (fraction of steps relative to the full chain-of-thought when prompting off-the-shelf LRMs in *thinking* mode). The $y$-axis shows the proportion of generated answers that match the model's own final answer (*Consistency*; left panel) or the proportion of generated answers that match the ground-truth answers (*Accuracy*; right panel) on MATH500. The early stop trend is constructed by prompting LRMs in *thinking* mode, splitting each response into steps, eliciting a predicted answer at each step, and plotting the fraction of steps against the proportion of matching answers across questions. The curve divides the plot into a top blue region (already matched) and a bottom green region (not yet matched). The red box shows the zone of optimal reasoning efficiency.

## 1 INTRODUCTION

Large reasoning models (LRMs) like OpenAI o1 and DeepSeek-R1 push the frontier of machine reasoning by generating long chains of thought (Li et al., 2025), achieving SOTA across medical QA, scientific reasoning, and math (Tang et al., 2025b; Rein et al., 2023; Hendrycks et al., 2021). A common way to boost accuracy is simply to allocate more reasoning tokens (Sheng et al., 2025; Yao et al., 2023; Tran et al., 2024), since longer thinking can self-correct via *self-verification* (He et al., 2025; Prystawski et al., 2023). Yet this also invites unnecessarily elaborate or redundant steps (Qu et al., 2025; Sui et al., 2025), inflating CoT length, slowing responses, and degrading user experience (Kumar et al., 2025).

---

[1]Our code: https://anonymous.4open.science/r/Shor-Think-1FC6/README.md

To improve reasoning efficiency, prior work has explored three main directions: explicit control of CoT length, adaptive control of CoT length, and incorporating the CoT length into the loss function. Methods for explicit control of the length of the CoT constrain the model's output length by, for instance, prompting a dummy thinking box so that the model outputs solutions directly Ma et al. (2025). Methods for adaptive control of the CoT length approach training LRMs to switch between two discrete modes of *Thinking* and *NoThinking* (no CoT), rewarding models for skipping reasoning on easier cases while maintaining accuracy on harder ones(Zhang et al., 2025b). Finally, some methods introduce a length penalty on the CoT length in on-policy reinforcement learning to generate shorter responses (Chang et al., 2025; Arora & Zanette, 2025).

Figure 2: **ESTAR can predict where to halt the CoT.** An illustration of (a) redundant reasoning, (b) standard efficient reasoning method, (c) our proposed redundant detection with LigthGBM. Green text: the necessary thinking steps—the portion of the reasoning where the model first arrives at the correct answer. Blue text: the redundant thinking steps—extra thinking generated after the correct answer has already been reached, often repetitive or even misleading. Purple text: the stop signal—a special token that indicates where the model should terminate its reasoning. Red text: the final answer—the model's selected output after completing its reasoning process.

However, the aforementioned methods overlook the observation that the optimal stopping point for the CoT may vary across data points and may lie midway through a reasoning trajectory rather than at its start or end. As shown in Figure 1, we examine the impact of the CoT length (reasoning progress) from two complementary perspectives: *consistency* (agreement with the model's own final prediction; left panel) and *accuracy* (agreement with the ground-truth answer; right panel). The black curves reveal that a large fraction (71%) of trajectories converge well before completion—intermediate predictions already match the final answer halfway through the process. Existing methods such as AdaptThink cannot reliably detect these early-stopping cutoffs for each data point. As shown in the example in Figure 2a, the LRM arrives at the correct answer after the necessary thinking steps (in green), yet the CoT continues generating additional tokens that introduce redundancy and even some temporary errors (in blue). Once the LRM enters the *thinking* mode, it typically includes both necessary as well as redundant steps. Whereas, switching to *NoThinking* could be less accurate (Figure 2b). By jointly examining accuracy and consistency in Figure 1, it becomes evident that consistency offers a reliable signal for identifying when reasoning can be safely truncated.

Building on the observation that reasoning trajectories often converge early, our work investigates the following research questions. **RQ1**: Can we develop a lightweight detector that reliably identifies when redundant thinking occurs, so that reasoning can be safely truncated without sacrificing accuracy? **RQ2**: Beyond detection, can LRMs themselves be trained to propose candidate stopping points, thereby narrowing the search space of potential early-stop positions? **RQ3**: By integrating self-generated stop signals with reinforcement learning, can we further improve efficiency while preserving task performance?

To address the above questions, first, we develop a lightweight method to detect when redundant thinking starts. We introduce early-stop step where reasoning is forcibly terminated and an answer is elicited. From a certain step, we extract features from LRM output, such as log-probability slopes, second-order differences, and stability signals (e.g., agreement across adjacent prefixes, flip counts). Using these features, we train a LightGBM classifier (ESTAR-LITE) to predict whether reasoning should stop or continue. Across four in-domain reasoning benchmarks (USMLE, JAMA, MATH500, AIME2025), ESTAR-LITE maintains at least 95% of the accuracy while shortening CoT by a factor of 2–6 compared to LRM in *thinking* mode. Moreover, on the out-of-domain GPQA dataset, ESTAR-LITE achieves similar results. It preserves $\geq 95\%$ accuracy and reducing reasoning length by 2–3 times, highlighting its strong cross-domain generalization.

Since detecting redundant thinking at each step is impractical, we design a post-training solution that allows LRMs to propose their own stopping points, which are then verified by ESTAR-LITE. We construct an SFT dataset by imposing fixed-length checkpoints and forcing an answer, labeling a step as positive only if the step's answer matches the full-think final answer. After ESTAR-FT, we adapt GRPO with a compute-aware reward that explicitly rewards correct `<stop>` emissions. During training, we truncate rollouts at `<stop>` and update ESTAR-LITE to stay aligned with the new trajectories. This yields our final model, ESTAR. On four reasoning benchmarks, ESTAR reduces average CoT length from 4799 to 1290 tokens (a $\times 3.7$ reduction) while retaining 98.9% of the original accuracy (74.9% $\rightarrow$ 74.2%). This outperforms other efficient reasoning baselines: LengthPenalty ($\times 1.4$ shorter, 97.0% relative accuracy) and AdaptThink ($\times 2.2$ shorter, 97.4% relative accuracy). In summary, ESTAR achieves substantial efficiency gains without compromising performance.

## 2 RELATED WORK

**Efficient reasoning through adaptive thinking** The simplest approach is to instruct the model to be brief or output fewer steps (Muennighoff et al., 2025; Ma et al., 2025). Another line of research trains models to adaptively decide between thinking or not thinking on a per-instance basis (Jiang et al., 2025b; Shen et al., 2025; Xu et al., 2025). For example, Zhang et al. (2025b) explicitly teaches a model when to think; it encourages the model to choose a direct answer (without *thinking*) for simpler queries while still using the complete CoT on harder queries. Similarly, Thinkless (Fang et al., 2025) trains a hybrid reasoner that outputs a special control token <short>, as the first output token, to decide the response style. Some methods have also explored routing strategies, such as using a separate classifier or heuristic to determine if a query needs CoT or not (Su et al., 2025; Marinelli et al., 2025). While binary mode-switching approaches are easy to train, they often require careful tuning to avoid overfitting (Yang et al., 2025b; Tang et al., 2025a; Jiang et al., 2025a; Zhang et al., 2025a; Wei et al., 2025). Our method differs from these binary mode-switching approaches by operating at a finer granularity: rather than an all-or-nothing decisions between reasoning and no-reasoning, we detect an early-stopping point within the CoT.

**Efficient reasoning with length-based penalties** Another line of work injects CoT length into the objective to directly optimize for brevity (Hou et al., 2025; Fatemi et al., 2025; Shrivastava et al., 2025; Lou et al., 2025; Dai et al., 2025). For instance, Chang et al. (2025) use a cosine length-scaling reward with a repetition penalty to suppress long chains during RL, curbing overthinking without hurting accuracy; Xiang et al. (2025) similarly penalize extra tokens once a correct answer has appeared. While such reward shaping globally biases models toward shorter outputs, it requires careful tuning and can induce under-thinking on hard cases (Yang et al., 2025c). In contrast, our proposed method directly uses the model's predicted stop point to truncate the CoT. Consequently, instead of just using a generic length penalty, we enforce brevity by halting generation at the detected early-stopping token. However, we also incorporate the length of the CoT into the loss function in order to reap the benefits of the above methods, while also using our early-stopping classifier.

## 3 PRELIMINARIES

**Problem setup.** Given a prompt $x = [x_1, \ldots, x_n]$, a reasoning policy $\pi_\theta$ produces a sequence

$$ y = \big[\texttt{<think>}, y_{1:\ell}, \texttt{</think>}, y_{\ell+1:m}\big], $$

where $y_{1:\ell}$ is the CoT and $y_{\ell+1:m}$ is the conclusion.

Let $\Omega$ be the complete answer set and $A(y) \in \Omega$ represents the answer extracted from $y$. For any prefix length $t \in \{0, \ldots, \ell\}$, we define a *forced early stop* that appends `</think>` after $y_{\leq t}$ and immediately elicits a conclusion, yielding a *early-stop answer*: $A_t^{\mathrm{ES}} = A\big([\texttt{<think>}, y_{\leq t}, \texttt{</think>}, \cdot]\big)$.

$t$ is a *safe stop* step if $A_t^{\mathrm{ES}} = A_\ell^{\mathrm{ES}}$; Our goal is to find the *earliest safe stop* $\tau^\star(x) = \min\{t \leq \ell : A_t^{\mathrm{ES}} = A_\ell^{\mathrm{ES}}\}$.

**Posterior viewpoint and safety certificate** Define the (implicit) answer posterior distribution if we stopped at $t$: $\vec{p}_t = \vec{\pi}_\theta\big(A_t^{\mathrm{ES}} \,\big|\, x, y_{\leq t}, \texttt{</think>}\big), A_t^{\mathrm{ES}} \in \Omega$.

Let $\mathcal{Y}_t = \sigma(y_{\leq t})$ be the distribution of $y_{\leq t}$, we can define *Tail Variation* as an expected accumulative distribution difference of early stop answers from $t$ to $\ell$:

$$\mathrm{TV}_t = \mathbb{E}\left[\sum_{k=t}^{\ell-1} \|\vec{p}_{k+1} - \vec{p}_k\|_1 \,\bigg|\, \mathcal{Y}_t\right],$$

it describes how much variance the final answer is expected if the LRM continues to think after step $t$.

Intuitively, a stop is safe when the remaining posterior movement is too small to overturn the current mode. This lead to the following theorum:

**Theorem 3.1.** *Let $\hat{A}_t = \arg\max_{A_t} p_t(A_t)$ and $\gamma_t = p_t(\hat{A}_t) - \max_{A_t \neq \hat{A}_t} p_t(A_t)$ be the confidence margin. Then, a sufficient (computable) stopping rule is $\tau^\dagger = \inf\{t : \mathrm{TV}_t \leq c\,\gamma_t\}$, where $c$ is a small positive scalar.*

## 4 METHODS

We address the challenge of redundancy in CoT by introducing into ESTAR. We address (**RQ1**) by building a lightweight *instance-wise early-stopping detector* which we refer to as ESTAR-LITE. ESTAR-LITE decides when to early stop a CoT by using a tabular classifier with features derived from the token probabilities. To answer (**RQ2**), we introduce ESTAR-F(INE)T(UNING) wherein the LRM undergoes supervised fine-tuning on curated CoT enabling the model to emit a special `<stop>` token that narrows down the search space of candidate early-stopping tokens. Finally, to answer (**RQ3**), we introduce ESTAR which augments ESTAR-FT with a reinforcement learning loss function featuring the length of the CoT. At inference time, earlystop is applied if the LRM proposes a `<stop>` token during its reasoning process and the tabular classifier also confirms the consistency of the model's earlystop answer.

### 4.1 CLASSIFIER PREDICTION

Based on Appendix §A.2, our stopping test depends on the tail variation of the answer posterior:

$$\mathrm{TV}_t \lesssim c_1 \sum_{k \in \mathcal{W}_t} |S_k| + c_2 \sum_{k \in \mathcal{W}_t} |H_k| + \mathcal{R}_t(w), \tag{1}$$

Here $S_t$ is the one-step marginal gain and $H_t$ is the discrete curvature (plateau indicator) of $p$.

However, converting this into a closed-form is infeasible because any series expansion would rely on unknown smoothness and conditioning constants, as well as a remainder term that is difficult to bound or estimate from the data. Consequently, rather than hard-coding thresholds, we operationalize Eq. (1) by extracting all terms it implicates, into observable proxies and learn a surrogate decision function that maps these proxies to stop or continue. Concretely, we instantiate four feature groups that cover the drivers in Eq. (1): (i) local dynamics of $L_t^{\mathrm{ES}}$ (slope and curvature) to capture marginal utility and saturation, (ii) cumulative margin and short-horizon stability to reflect argmax robustness, (iii) instantaneous evidence to absorb remainder terms and class-redistribution effects, and (iv) a progress ratio as a weak prior on budget usage. We then fit a nonparametric classifier (LightGBM) to these features, allowing piecewise decision surfaces that accommodate heterogeneous smoothness and higher-order interactions, thereby replacing brittle hand-tuned thresholds with a data-driven estimator of the stopping boundary. Additional details about how to calculate the features are described in Appendix A.3. Below, we describe the intuition behind the set of features used in ESTAR-LITE.

**(1) Instantaneous evidence.** At step $t$, the model induces a next-token distribution $\pi_\theta(\text{tok} \mid s_{<t})$. We map each token to an answer bucket $\Omega$ (e.g., canonicalizing token surface forms; for multiple-choice tasks this reduces to a letter–option map). For each bucket $\omega \in \Omega$, we aggregate token-level evidence by a log-sum-exp over tokensassigned to $\omega$ (denoted as $\text{inst\_s}(\omega, t)$), and then normalize across $\Omega$ to obtain a per-class instantaneous probability $\text{inst\_p}(\omega, t)$. Intuitively, $\text{inst\_p}$ summarizes the model's current local preference over answers before consuming the next token.

**(2) Cumulative path & stability.** We maintain a class-wise cumulative evidence vector $C(t)$ whose $\omega$-th component accumulates $\log \text{inst\_p}(\omega, t)$ from $t = 1$ to now. The running winner (the $\arg\max_\omega C_\omega(t)$) defines the current predicted answer. From this path we derive several stability cues: *flips(t)* counts the number of winning changes so far, and *changed_prev(t)* indicates whether token $t$ introduced a new winner. Together they measure the stickiness of the decision and the volatility of the trajectory.

**(3) Early-stop curvature cues.** Let $L_t^{\text{ES}}$ denote an early-stopping confidence score (higher is more confident to stop). We use its one-step slope $\text{S\_es}(t)$ to proxy marginal gain from one extra token, and its second difference $\text{H\_es}(t)$ to indicate local saturation or acceleration.

**(4) Token-level confidence statistics.** From the per-step token log probabilities inside the predicted answer span, we compute the mean $\mu_t$ and variance $\sigma_t^2$ (confidence and dispersion), a negative-perplexity proxy $\text{neg\_ppl}(t)$ (higher means more peaked), and the token length $\text{ans\_len(t)}$ (longer spans often correlate with lower certainty).

**(5) Classifier.** Concatenating these signals yields a stepwise feature vector $\phi_t$. We train a classifier $C_\varphi(\phi_t) \in [0, 1]$ with a logistic loss, where $y_t \in 0, 1$ marks whether stopping at $t$ preserves the final answer. A tree ensemble converts curvature, stability, and evidence cues into a calibrated stop probability—data-driven and nonparametric, without hand-tuned thresholds.

**ESTAR-LITE during inference.** During inference we update $\phi_t$ online and evaluate $\rho_t = C_\varphi(\phi_t)$ at each step. We stop at the earliest $t$ satisfying $\rho_t \geq \tau$ ($\tau$ is set to 0.9 throughout this manuscript) and a short patience check (e.g., whether the condition holds for a few consecutive steps), then append the `</think>` token and elicit the final answer.

## 4.2 SELF-GENERATED STOP CUE VIA SFT

When applying the above classifier during inference, it requires a pre-determined task-wise step size during generation (e.g. classifies every 10 CoT tokens). Such *global* strategy could lead to sub-optimal early stopping (as shown in Sec. 6.2). In this section, our goal is use a similar training signals from above to teach the LRM to propose early-stop positions by itself. First, we insert `<stop>` tokens into CoTs at the position where early-stop labels are positive. Then we apply regular teacher forcing supervised finetuning on these formatted CoTs. In particular, given a data instance $(x, y)$, we first segment its CoT into sentences $[s_1..., s_M]$. For each prefix $(x, s_{1:j}|_{j=1}^M)$ we conduct forced early stop and let the LRM directly generate $A^{ES}$; If $A^{\text{ES}}$ equals $A_\ell^{\text{ES}}$, we insert a `<stop>` token at $s_j$. The `<stop>`-inserted $(x, y')$ forms the training data for SFT.

Two additional enhancements are made to create the training signal. Firstly, to encourage earlier exits, we add at most first 5 `<stop>` tokens per instance. Secondly, to learn from more challenging data, we apply inference scaling ( Muennighoff et al. (2025)) when CoTs lead to incorrect answers. Concretely, given an instance $(x, y)$ where answer is incorrect, we form a new prompt with a hint containing gold answer $A^\star$: $x_c = [x, \texttt{<think>}y_{1:\ell} \texttt{</think>}, h]$ where $h = $ `"Wait the correct answer is $A^\star$ "` and let LRM complete the rest of the reasoning trace. This scaling approach forms an corrected thinking trace $y_c$ that can lead to the gold answer. Then we insert `<stop>` tokens into $y_c$ using the same approach described above. This augmentation preserves the original reasoning prefix while supplying positive supervision even when the initial trajectory fails. Afterwards, we conduct supervised finetuning on the collected training data with the objective of minimizing the cross entropy loss for next word prediction.

## 4.3 STOP-AWARE REINFORCEMENT LEARNING

Supervised finetuning a LLM on off-policy reference data often leads to memorization and may harm its generalization ability (Chu et al.). To tackle RQ3, we convert the early stop signals as a reward

function in reinforcement learning and the objective is to encourage the LRM to generate earliest possible `<stop>` tokens that trigger valid solutions.

**Policy rollout.** For a prompt $x$, the policy $\pi_\theta$ generates a whole-thinking sequence $y_{1:\ell_{\text{gen}}}$ which may contain a set of early stop proposals:

$$\mathcal{P}(x) \;=\; \{\, t \le \ell_{\text{gen}} : y_t = \texttt{<stop>} \,\}.$$

For all $t \in \mathcal{P}(x)$, we force a conclusion and test acceptance:

$$A_t^{\text{ES}} \sim \pi_\theta(\cdot \mid x, y_{\le t}, \texttt{</think>}), \qquad a_t \;=\; \mathbf{1}\{\, A_t^{\text{ES}} = A^\star \,\},$$

The rollout is truncated to the earliest safe stop:

$$\tilde{t} \;=\; \min\{\, t \in \mathcal{P}(x) : a_t = 1 \,\} \qquad \tilde{\tau} \;=\; \big( y_{1:\tilde{t}}, \, A_{\tilde{t}}^{\text{ES}} \big).$$

If $a_t = 0$ for all $t \in \mathcal{P}(x)$), we set $\tilde{t} = \ell_{gen}$ for credit assignment. This verify-then-truncate protocol ensures that only a verified proposal terminates the rollout; Earlier proposals that fail verification do not trigger truncation and therefore receive rewards.

**Reward.** Each proposal before truncation $t' \in \{t \in \mathcal{P}(x) \wedge t \le \tilde{t}\}$ receives the following reward:

$$r(t') \;=\; \lambda_1 r_{\text{fmt}}(t') \;+\; \lambda_2 r_{\text{stop}}(t') \;+\; \lambda_3 r_{\text{acc}}\big(A_{t'}^{\text{ES}}, A^\star\big). \tag{2}$$

Here $r_{\text{fmt}}$ enforces well-formed use/order of `<think>`, `<stop>`, `</think>`; $r_{\text{stop}}$ is monotone in earliness (e.g., decreasing in $\tilde{t}$ or a normalized progress ratio) and is granted only when $t' = \tilde{t}$; $r_{\text{acc}}$ scores the final answer against the gold answer. We use GRPO(Shao et al., 2024) in our reinforcement learning process due to its efficient learning with rewards from multiple rollouts.

**Inference.** At test time we combine the model's self-generated `<stop>` cue with an external classifier to decide termination. During inference, whenever a `<stop>` token is emitted at step $t$, we treat $t$ as an early stop proposal. We then compute the feature vector $\phi_t$ online and the classifier produces $\rho_t = C_\varphi(\phi_t)$. If $\rho_t \ge \tau$ (optionally after a brief patience check), we accept the stop: append `</think>`, elicit the final answer; otherwise, the model continues the generation before `</think>` until a new stop token is proposed or a budget is reached.

## 5 EXPERIMENTS

### 5.1 EXPERIMENTAL SETUP

**Datasets.** We evaluate on five public benchmarks USMLE-QA (Jin et al., 2020) and JAMA-QA (Chen et al., 2025) are multiple-choice medical QA (closed-form answers). GPQA-Diamond (Rein et al., 2023) is a hard STEM multiple-choice set (closed-form answers). Math-500 (Satpute et al., 2024) and AIME2025 are open-ended math sets (short textual/numeric answers).

### 5.2 RQ1: ROBUSTNESS AND LOW-COST PREDICTION EXPERIMENT SETTINGS

**ESTAR-LITE: Training a lightweight classifier to determine early stopping.** We use Qwen3-8B and Qwen3-14B (Yang et al., 2025a). To train the classifier, we first generate the chain-of-thought (CoT) for each data point with temperature 0.6, top-$k$=20, top-$p$=0.95, repetition penalty 1.2.

We then uniformly slice the CoT at $10\%, 20\%, \ldots, 100\%$ of its length (in number of tokens) and append the `</think>` token to elicit a conclusion with greedy decoding (temperature 0). At each token $t$ of the CoT, we log top-20 next-token log-probabilities, bucket tokens into the answer set $\Omega$, and compute the online features $\phi_t$.

We train a LightGBM classifier $C_\varphi$ (using a binary cross-entropy loss function) on pairs $(\phi_t, y_t)$ from each token $t$ of the CoT using the default hyperparameters: n_estimators=400, num_leaves=63, learning_rate=0.07, subsample=0.9, colsample_bytree=0.9. To train LightGBM for Closed QA task, we used train data split of USMLE-QA. To train LightGBM for Open QA task, we used DEEPSCALER (Luo et al., 2025). For in-domain evaluation on Open QA tasks, we used MATH500 and AIME. For in-domain evaluation on Closed QA tasks, we used JAMA-QA and test split of USMLE-QA. For out-of-domain evaluation, we used GPQA.

| Model | Dataset | Accuracy | | Length | |
|-------|---------|-------------|------------|-------------|------------|
| | | Traditional | ESTAR-LITE | Traditional | ESTAR-LITE |
| | USMLE | 77.53 | 76.83 (99.1%) | 2412 | 549 ($\times$4.4) |
| | JAMA | 57.45 | 56.55 (98.4%) | 2423 | 419 ($\times$5.8) |
| Qwen3-8B | GPQA | 60.10 | 59.50 (99.0%) | 3882 | 1695 ($\times$2.3) |
| | MATH500 | 94.00 | 93.20 (99.2%) | 3951 | 2019 ($\times$2.0) |
| | AIME | 70.00 | 66.67 (95.2%) | 6123 | 3045 ($\times$2.0) |
| | USMLE | 77.83 | 77.04 (99.0%) | 1985 | 302 ($\times$6.6) |
| | JAMA | 59.76 | 59.89 (100.2%) | 2047 | 438 ($\times$4.7) |
| Qwen3-14B | GPQA | 66.67 | 66.67 (100.0%) | 4312 | 1161 ($\times$3.7) |
| | MATH500 | 94.00 | 93.20 (99.1%) | 3525 | 1314 ($\times$2.7) |
| | AIME | 70.00 | 66.67 (95.2%) | 5789 | 2891 ($\times$2.0) |

Table 1: Accuracy and CoT length on LRMs with *thinking* mode (Traditional) vs. early-stopping (ESTAR-LITE). Parentheses show relative values. ESTAR-LITE typically preserves $\geq 99\%$ accuracy while reducing reasoning length by $\times$2–6.

## 5.3 RQ2: LEARNING <stop> PROPOSALS TO REDUCE EARLY-STOPPING CANDIDATES

**ESTAR-FT: Supervised fine-tuning LRMs to introduce the ability to emit `<stop>` tokens.** To introduce the capability of the LRM to self-propose `<stop>` tokens, we generate supervised fine-tuning data on MATH-500. Concretely, for each data point we generate the CoT $[y_1, \ldots, y_\ell]$ followed by a final answer. We fine-tune with token-level cross-entropy (learning rate = $2\times10^{-5}$; 8 GPUs), treating the `<stop>` token as part of the vocabulary so that the LRM learns to emit `<stop>` tokens.

## 5.4 RQ3: CORRECTNESS-VERIFIED <stop> UNDER RL CAN IMPROVE THINKING EFFICIENCY

**ESTAR: Reinforcement learning with ESTAR-FT and a length penalty.** We modify the traditional reinforcement learning using GRPO by adding a length penalty in the reward function using $\lambda_1 = 1/4, \lambda_2 = 1/4, \lambda_3 = 1/2$ in equation 2. During training, when the policy emits a `<stop>` token at step $t$, we append a `</think>` token to obtain $A_t^{\mathrm{ES}}$. If $A_t^{\mathrm{ES}} = A^\star$, then we truncate the roll-out at token $t$. We trained batch size 256, $n_{\text{roll-outs}} = 16$, for 39 steps. For close-ended tasks, we run GRPO on USMLE-QA–Train. For open-ended tasks, we run GRPO on deepscaleR.

During inference with ESTAR, when the LRM emits a `<stop>` token at step $t$ within its CoT, the classifier compute a logit $\rho_t = C_\varphi(\phi_t)$. If $\rho_t \geq \tau$ (we set $\tau = 0.9$ across all experiments presented in this manuscript), we append the `</think>` token, elicit the answer, and early-stop; otherwise we continue until the next `<stop>` or the CoT terminates.

## 6 RESULTS

### 6.1 RQ1: ESTAR-LITE: A LIGHTWEIGHT CLASSIFIER TO IDENTIFY WHEN TO EARLY-STOP

**ESTAR-LITE leads to efficient inference without sacrificing accuracy** Table 1 shows that the lightweight classifier within ESTAR-LITE (using a constant threshold $\tau = 0.9$ across datasets) leads to a $\geq \times4$ improvement in efficiency while preserving accuracy to within $\geq 95\%$ of that obtained with the full CoT on the medical QA USMLE dataset. We observe a similar pattern in JAMA. For the hard STEM dataset, GPQA, we still observe $\geq \times2$ improvement in efficiency while preserving accuracy to within $\geq 99\%$ of that obtained with the full CoT. In the open-ended math dataset, MATH500, we see a similar pattern. Additionally, in the open-ended math dataset, AIME, we still observe $\geq \times2$ improvement in efficiency while preserving accuracy to within $\geq 95\%$ of that obtained with the full CoT. These results indicate that ESTAR-LITE is able to generalize to out-of-domain datasets since ESTAR-LITE was trained only on USMLE-QA-train and deepscaleR and its classification threshold remains constant across datasets ($\tau = 0.9$). See Appendix A.5 for additional analysis on how the ESTAR-LITE adapts reasoning length to question difficulty.

**ESTAR-LITE features are discriminative and easy to compute** Figure 3 demonstrates that two of the ESTAR-LITE features, which are easy to compute, `slope_recent` and `delta_recent`, cleanly separate CoT steps whose answer already matches the final answer (*match*) from those that do not

(*mismatch*). Because these features are model-agnostic and require access to only the top-$k$ token log-probabilities, they are inexpensive to compute during inference.

**ESTAR-LITE classification threshold trades off accuracy vs coverage**   Table Suppl. 1 summarizes the tradeoff between accuracy, defined as the percentage of datapoints for which the early-stopped answer matches the answer derived using the full CoT, and the coverage, defined as the percentage of data points for which ESTAR-LITE led to early-stopping. We observe that as the ESTAR-LITE classification threshold, ($\tau$), decreases, the accuracy as well as the length of the early-stopped CoT goes down, while the coverage increases. Since the accuracy remains stable across a range of $\tau$ values, we set $\tau = 0.9$ for all analyses presented in this manuscript.

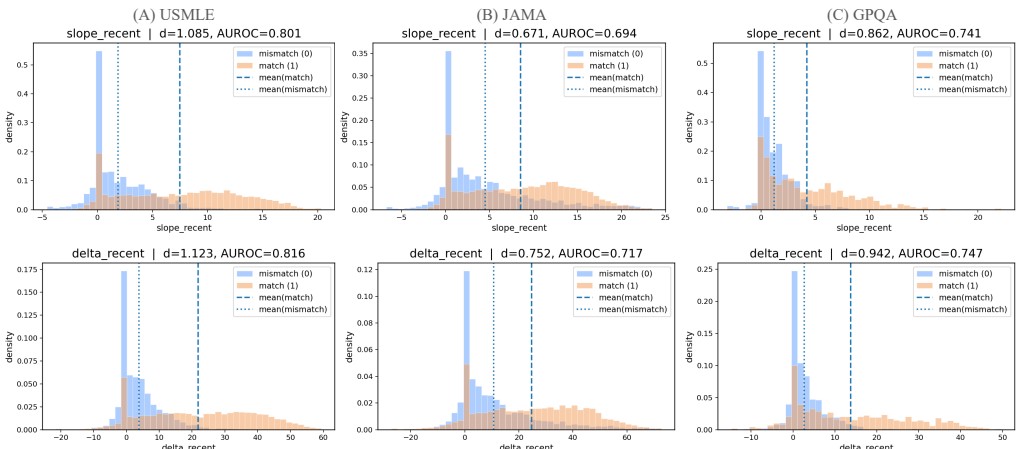

Figure 3: **ESTAR-LITE features separate tokens where the answer matches the one with full-CoT.** Each panel shows a density–normalized histogram of a feature for two classes: match (the token's answer equals the full-CoT answer) and mismatch (where it does not). Top row: slope_recent; bottom row: delta_recent. Vertical dashed lines mark class means (long dash = match; dotted = mismatch). Panel titles report Cohen's $d$ and AUROC, computed on all tokens for that dataset, using the feature value as the score (higher $\Rightarrow$ more likely match). AUROC is the area under the ROC curve, i.e., the probability that a randomly chosen match step receives a higher feature value than a randomly chosen mismatch step ($0.5$ = random guessing; $1.0$ = perfect).

### 6.2   RQ2: ESTAR-FT: LEARNING `<stop>` PROPOSALS TO PRUNE EARLY-STOP CANDIDATES

On MATH-500, if we vary the frequency with which the ESTAR-LITE classifier is invoked, we get varying points along the Consistency vs Reasoning length plot shown in Figure 4. Among the markers corresponding to ESTAR-LITE, we vary invoking the classifier every 20, 40, ..., 200 tokens, where the lightest marker corresponds to invoking the classifier every 20 tokens. We observe that invoking the classifier more frequently (lighter markers) lead to shorter reasoning length (represented by the x-axis) with a marginal drop in consistency (represented by the y-axis).

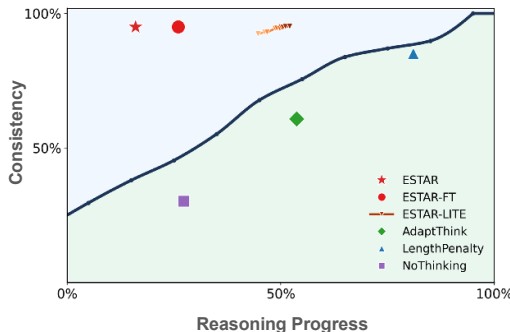

Figure 4: ESTAR and ESTAR-FT stop earlier, achieving high consistency with fewer checks.

ESTAR-FT presents an alternative to the strategy of tuning this frequency of invoking the ESTAR-LITE classifier. In ESTAR-FT, since the LRM is fine-tuned to reliably emit `<stop>` tokens, its consistency is better than the best choice the frequency of ESTAR-LITE while having a shorter early-stopped CoT length. ESTAR-FT is therefore able to adaptively determine when to emit a `<stop>`

| Method | Closed QA | | | | Open QA | | | |
| | USMLE | | JAMA | | MATH500 | | AIME | |
| | Acc | Len | Acc | Len | Acc | Len | Acc | Len |
|---|---|---|---|---|---|---|---|---|
| GRPO | 78.14 | 2732 | 57.8 | 2634 | 94.0 | 3962 | 70.00 | 9871 |
| | (100.0%) | (x1.0) | (100.0%) | (x1.0) | (100.0%) | (x1.0) | (100.0%) | (x1.0) |
| No-Thinking | 66.2 | 315 | 48.2 | 369 | 85.0 | 1139 | 26.67 | 1513 |
| | (84.7%) | (x8.7) | (83.4%) | (x7.1) | (90.4%) | (x3.5) | (38.1%) | (x6.5) |
| AdaptThink | 76.4 | 987 | 55.8 | 1102 | 93.8 | 2130 | 66.67 | 4513 |
| | (97.8%) | (x2.8) | (96.5%) | (x2.4) | (99.8%) | (x1.9) | (95.2%) | (x2.2) |
| Length-Penalty | 76.6 | 1325 | 55.4 | 1872 | 92.4 | 3190 | 66.67 | 7324 |
| | (98.0%) | (x2.1) | (95.8%) | (x1.4) | (98.3%) | (x1.2) | (95.2%) | (x1.3) |
| ESTAR-LITE | 76.83 | 549 | 56.55 | 419 | 93.2 | 2019 | 66.67 | 3045 |
| | (98.3%) | (x5.0) | (97.8%) | (x6.3) | (99.1%) | (x2.0) | (95.2%) | (x3.2) |
| ESTAR | 77.13 | 388 | 56.10 | 352 | 93.8 | 635 | 70.00 | 3788 |
| | (98.7%) | (x7.0) | (97.1%) | (x7.5) | (99.8%) | (x6.2) | (100.0%) | (x2.6) |

Table 2: Accuracy / length by method, grouped by task type. **Closed QA** (USMLE, JAMA) has a single correct option; **Open QA** (MATH500, AIME2025) requires free-form numeric/algebraic answers. For each dataset, Acc is test accuracy (in %), and Len is the average output token length of the response (including reasoning, if any).

token and thereby reduce the CoT length without sacrificing accuracy. LRMs can thus be supervised fine-tuned to propose accurate early-stopping tokens that improve the tradeoff between accuracy and number of invocations to the ESTAR-LITE classifier.

### 6.3 RQ3: ESTAR: CORRECTNESS-VERIFIED `<stop>` UNDER RL CAN IMPROVE THINKING EFFICIENCY

We compare methods on two families of tasks: *Closed QA* (USMLE, JAMA; single correct option) and *Open QA* (MATH500, AIME2025; free-form numeric/algebraic answers) as shown in Table 2. For each dataset, **Acc** represents the test accuracy (%) and **Len** represents the average number of output tokens including the (early-stopped) CoT.

Across all datasets, **ESTAR** achieves $> 97\%$ accuracy as compared to reinforcement learning using GRPO with *substantially shorter* CoTs. We observe an increase (in terms of length of early-stopped CoT) of $\sim 7\times$ on Closed QA and by $\sim 6\times$ on MATH500. On the AIME dataset, we observe no reduction in accuracy (70.0% vs. 70.0%) with $\sim 2.6\times$ fewer tokens in the early-stopped CoT obtained with ESTAR. Compared with AdaptThink, ESTAR attains a superior accuracy-efficiency tradeoff. For instance, on the MATH500 dataset, it matches AdaptThink's 93.8% with $\approx 70\%$ fewer tokens (635 vs. 2130). Similarly, on the USMLE dataset, ESTAR achieves 77.1% accuracy (vs. 76.4%) with $\approx 61\%$ fewer tokens (388 vs. 987). Introducing a length penalty in the reward for reinforcement learning shortens the CoT but not as much as in ESTAR with bigger drops in accuracy. No-Thinking, by construction, leads to the shortest CoTs; however the consequent accuracy suffers large drops across datasets.

## 7 CONCLUSION

Strategies for making inference with large reasoning models more efficient, such as global penalties or coarse mode switches (e.g., introducing a length penalty, AdaptThink etc.) cannot detect mid-trajectory convergence and thus waste computation. In this manuscript, we introduce ESTAR which combines a lighweight classifier to predict consistency with the answer derived using a full CoT and per-instance proposals to train a policy to act on them only when correctness is verified. This turns an exhaustive scan over all sentence boundaries into validating a small set of candidates. This targeted truncation consistently delivers $\times 4$–$10$ CoT length reductions (and therefore efficiency gains) with essentially unchanged accuracy, demonstrating that efficiency gains need not trade off performance. Our results establish a path toward practical reasoning systems that are both accurate and compute-efficient.

## 8 REPRODUCIBILITY STATEMENT

We have taken several steps to ensure the reproducibility of our work. All models and algorithms are described in detail in the main text (Sections 3 and 4), with theoretical formulations of the motivation provided in Appendix A. The description of datasets and preprocessing steps is given in Section 5.1. Hyperparameters and training configurations are reported in Section 5.2. We also shared our code link in page 1.

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

# A APPENDIX

## A.1 LLM USAGE

In accordance with the ICLR 2026 policies on LLM usage, we disclose how LLMs were used in this work. LLMs were employed to assist with grammar polishing, wording improvements, and drafting text during paper preparation. All technical content, proofs, experiments, and analyses were conceived, implemented, and validated by the authors. Authors remain fully responsible for the correctness of the claims and results.

No LLMs were used to generate research ideas or produce results. No confidential information was shared with LLMs, and no prompt injections or other inappropriate uses were involved.

This disclosure aligns with the ICLR Code of Ethics: contributions of tools are acknowledged, while accountability and verification rest entirely with the human authors.

## A.2 MOTIVATION

**Generative setup.** Consider a reasoning model $\pi_\theta$ that, given a prompt $x = [x_1, \ldots, x_n]$, produces

$$y = \big[ \texttt{<think>}, y_1, \ldots, y_\ell, \texttt{</think>}, y_{\ell+2}, \ldots, y_m \big].$$

The segment $[y_1, \ldots, y_\ell]$ is the whole-thinking portion; $\texttt{</think>}$ marks the transition to the conclusion $[y_{\ell+2}, \ldots, y_m]$. The sequence $y$ is sampled from $\pi_\theta(\cdot \mid x)$ with the standard autoregressive factorization

$$\pi_\theta(y \mid x) = \prod_{t=1}^{m} \pi_\theta\big(y_t \mid x, y_{<t}\big).$$

Let $\Omega$ be the answer set and let $A(y) \in \Omega$ denote the answer extracted from the conclusion. For any thinking length $t \in \{0, \ldots, \ell\}$, define a forced early stop that truncates at $t$ by appending $\texttt{</think>}$ and immediately eliciting a conclusion; denote the induced answer by

$$A_t^{\mathrm{ES}} = A\big([\texttt{<think>}, y_{\leq t}, \texttt{</think>}, \cdot]\big).$$

A prefix $t$ is a safe stopping point if $A_t^{\mathrm{ES}} = A_\ell^{\mathrm{ES}}$; the *earliest safe stop* is

$$\tau^\star(x) = \min\{ t \leq \ell : A_t^{\mathrm{ES}} = A_\ell^{\mathrm{ES}} \}.$$

**Finite-horizon optimal stopping.** Thinking has a finite horizon. Let $\ell(x)$ be the number of whole-thinking tokens before $\texttt{</think>}$, write the thinking trace as $y_{1:\ell} = (y_1, \ldots, y_\ell)$, and let the observed prefix be $y_{\leq t} = (y_1, \ldots, y_t)$ with information $\mathcal{Y}_t = \sigma(y_{\leq t})$. Define the answer posterior the model would use if we stopped at $t$:

$$\vec{p}_t = \vec{\pi}_\theta\big(A_t^{\mathrm{ES}} \mid x, y_{\leq t}, \texttt{</think>}\big), A_t^{\mathrm{ES}} \in \Omega.$$

So we could get:

$$\|\vec{p}_\ell - \vec{p}_t\|_1 \;\leq\; \sum_{k=t}^{\ell-1} \|\vec{p}_{k+1} - \vec{p}_k\|_1.$$

where $\ell = \ell(x)$ denote the number of whole-thinking tokens produced before the delimiter `</think>` for instance $x$ (so the thinking trace is $s_{1:\ell}$). Define *tail variation*

$$\mathrm{TV}_t \;=\; \mathbb{E}\!\left[\sum_{k=t}^{\ell-1} \|p_{k+1} - p_k\|_1 \;\middle|\; \mathcal{Y}_t\right],$$

which necessarily satisfies $\mathrm{TV}_t \downarrow 0$ as $t \uparrow \ell$. Let $\hat{A}_t = \arg\max_{A_t} p_t(A_t)$ and

$$\gamma_t = p_t(\hat{A}_t) - \max_{A_t \neq \hat{A}_t} p_t(A_t)$$

be the confidence margin. A sufficient safety condition is that the tail cannot overturn the current prediction, e.g.

$$\mathrm{TV}_t \;\leq\; c\,\gamma_t,$$

for a small constant $c$ reflecting the Lipschitz sensitivity of the argmax map.[2]

We use the following tractable sufficient rule as a computable certificate:

$$\tau^\dagger \;=\; \inf\{\, t :\; \mathrm{TV}_t \leq c\,\gamma_t \,\},$$

i.e., stop when the remaining room for posterior change is too small to overturn the current decision.

**Why exact evaluation is hard.**  Both $\mathrm{TV}_t$ and $\gamma_t$ depend on the *future* posterior path $\{p_k\}_{k>t}$ and the continuation policy, so computing them online would require simulating counterfactual continuations—prohibitively expensive at decode time. Moreover, $p_t$ is implicit in hidden states and not directly observable.

**Theorem A.1** (**From optimal stopping to curvature-based proxies.**). *At each prefix $t$ we (conceptually) force* `</think>` *and define the early-stop confidence*

$$L_t^{\mathrm{ES}} = \log \pi_\theta\big(A_t^{\mathrm{ES}} \mid x, s_{\leq t}, \texttt{</think>}\big)$$

*Assuming mild smoothness of the path $t \mapsto L_t^{\mathrm{ES}}$, a local second-order finite-difference expansion along token time yields*

$$L_{t+1}^{\mathrm{ES}} \;\approx\; L_t^{\mathrm{ES}} \;+\; S_t \;+\; \tfrac{1}{2}\,H_t, \qquad S_t := L_t^{\mathrm{ES}} - L_{t-1}^{\mathrm{ES}}, \quad H_t := L_t^{\mathrm{ES}} - 2L_{t-1}^{\mathrm{ES}} + L_{t-2}^{\mathrm{ES}}.$$

*Here $S_t$ is the one-step marginal gain and $H_t$ the discrete curvature (plateau indicator). One-step posterior drift satisfies $\|p_{t+1} - p_t\|_1 \leq \sqrt{2\,\mathrm{KL}(p_{t+1} \,\|\, p_t)}$; while the KL is unobserved, small $|S_t|$ and near-zero $|H_t|$ indicate that $L_t^{\mathrm{ES}}$ has entered a local plateau, making further drift—and thus tail variation—negligible.*

$$\mathrm{TV}_t \;\lesssim\; c_1 \sum_{k \in \mathcal{W}_t} \big|S_k\big| \;+\; c_2 \sum_{k \in \mathcal{W}_t} \big|H_k\big| \;+\; \mathcal{R}_t(w),$$

### A.3   Features for Classifier

**Feature design and notation (complete).**  We work with a stepwise CoT trace. At each step $t = 1, 2, \ldots$ the model has history $s_{<t}$ and produces an *answer fragment*. We denote the model's next-token distribution by $\pi_\theta(\mathrm{tok} \mid s_{<t})$ over the vocabulary $\mathcal{V}$.

**Answer buckets.**  Let $\Omega$ be the finite set of answer "buckets" (e.g., $\{A, B, C, D\}$; for open-form tasks $\Omega$ can be a canonical-expression set).

---

[2]If the posterior can move by at most $\mathrm{TV}_t$ and the lead is $\gamma_t$, the predicted answer is stable.

**(1) Instantaneous evidence.** We aggregate token log-probabilities at step $t$ by bucket:

$$\texttt{inst\_s}(\omega, t) = \text{LSE}\Big\{ \log \pi_\theta(w \mid s_{<t})\Big\}, \omega \in \Omega \tag{3}$$

$$\texttt{inst\_p}(\omega, t) = \frac{\exp\big(\texttt{inst\_s}(\omega, t)\big)}{\sum_{\omega' \in \Omega} \exp\big(\texttt{inst\_s}(\omega', t)\big)}. \tag{4}$$

*Definitions used here:* $\texttt{inst\_s}(\omega, t)$ is the bucket-level log-evidence (log-sum-exp over tokens mapped to $\omega$); $\texttt{inst\_p}(\omega, t)$ is the normalized instantaneous probability over $\Omega$.

**(2) Cumulative path & stability.** From instantaneous probabilities we maintain class-wise running evidences

$$C_\omega(t) = C_\omega(t-1) + \log \texttt{inst\_p}(\omega, t), \quad C_\omega(0) = 0, \tag{5}$$

and the current winner $\hat{\omega}_t = \arg\max_\omega C_\omega(t)$ with margin

$$\Delta_t = C_{\hat{\omega}_t}(t) - \max_{\omega \neq \hat{\omega}_t} C_\omega(t). \tag{6}$$

We expose stability indicators

$$\texttt{run\_len}(t) = \Big|\{\tau \leq t : \hat{\omega}_\tau = \hat{\omega}_t\}\Big|, \qquad \texttt{flips}(t) = \sum_{\tau=2}^{t} \mathbf{1}\{\hat{\omega}_\tau \neq \hat{\omega}_{\tau-1}\}, \tag{7}$$

$$\texttt{changed\_prev}(t) = \mathbf{1}\{\hat{\omega}_t \neq \hat{\omega}_{t-1}\}. \tag{8}$$

*Meanings:* $\Delta_t$ is the winner's log-margin; $\texttt{run\_len}$ is the current run length of the winner; $\texttt{flips}$ counts winner changes up to $t$; $\texttt{changed\_prev}$ is the most recent-change flag.

**(3) Early-stop curvature cues.** Let $L_t^{\text{ES}}$ be the *early-stop confidence* instantiated at step $t$ as the log-probability *sum over the answer tokens* generated at step $t$ (i.e., the sum of token log-probs inside the first closing brace of \boxed{·}). We model its short-horizon kinematics:

$$\texttt{S\_es}(t) = L_t^{\text{ES}} - L_{t-1}^{\text{ES}} \qquad \text{(discrete slope)}, \tag{9}$$

$$\texttt{H\_es}(t) = \texttt{S\_es}(t) - \texttt{S\_es}(t-1) \qquad \text{(second difference)}, \tag{10}$$

$$\tag{11}$$

We further fit a quadratic on a sliding window of length $W$:

$$L_\tau^{\text{ES}} \approx a\,\tau^2 + b\,\tau + c, \qquad \tau \in \{t - W + 1, \dots, t\},$$

*Hyperparameters used in practice:* short-horizon size $K \in \{3, \dots\}$, quadratic window $W \in \{5, \dots\}$, anchor index $t_0 \in [t - W + 1, t]$ (we use the window start). All finite differences use zero when the needed past terms are absent (e.g., at the beginning of a trajectory).

**(4) Answer-token statistics.** Let the step-level answer span at step $t$ contain $n_t$ tokens with log-probabilities $\{\ell_i\}_{i=1}^{n_t}$. We compute

$$\mu_t = \frac{1}{n_t}\sum_{i=1}^{n_t} \ell_i, \qquad \sigma_t^2 = \frac{1}{n_t}\sum_{i=1}^{n_t}(\ell_i - \mu_t)^2, \qquad \texttt{neg\_ppl}(t) = -\mu_t, \qquad \texttt{ans\_len}(t) = n_t. \tag{12}$$

*Fallback:* if per-token logs are not available, $\mu_t$ backs off to the exported average log-probability and $n_t$ to the exported answer length.

**(5) Feature vector and learning.** The stepwise feature aggregates stability, curvature/kinematics, and token statistics:

$$\phi_t = \left[ \begin{array}{l} \Delta_t,\ \texttt{run\_len}(t),\ \texttt{flips}(t),\ \texttt{changed\_prev}(t) \ \text{(stability)} \\ \texttt{S\_es}(t),\ \texttt{H\_es}(t),\ \text{(curvature/kinematics)} \\ \mu_t,\ \sigma_t^2,\ \texttt{neg\_ppl}(t),\ \texttt{ans\_len}(t)\ \text{(token stats)} \end{array} \right].$$

explicitly excluding the absolute step ratio.

We learn a stop/continue classifier $C_\varphi : \phi_t \mapsto [0, 1]$ from labeled samples $(\phi_t, y_t)$ with the logistic objective

$$\min_\varphi \frac{1}{N}\sum_{t=1}^{N} \Big[ -y_t \log \sigma\big(f_\varphi(\phi_t)\big) - (1 - y_t) \log\big(1 - \sigma(f_\varphi(\phi_t))\big)\Big],$$

where $y_t = 1$ iff stopping at $t$ preserves the final answer (i.e., the winner at $t$ equals the final winner), $f_\varphi$ is the tree-ensemble score, and $\sigma$ is the sigmoid.

## A.4 ESTAR-LITE CLASSIFICATION THRESHOLD TRADES OFF

| Threshold ($\tau$) | Accuracy | Length | Coverage |
|---|---|---|---|
| 0.99 | 77.53% | 463 | 87.5% |
| 0.95 | 76.94% | 344 | 92.9% |
| 0.90 | 76.83% | 276 | 95.2% |
| 0.85 | 75.10% | 223 | 96.9% |
| 0.80 | 73.76% | 185 | 97.5% |

Table Suppl. 1: Tradeoff between Accuracy and early-stopped CoT Length as a function of the ESTAR-LITE classification threshold ($\tau$). Accuracy: fraction of data points where the early-stopped answer matches the ground-truth answer. Length: median CoT length with early-stopping using ESTAR-LITE. Coverage: percentage of data points for which the ESTAR-LITE led to early-stopping.

## A.5 ESTAR-LITE COT LENGTH AND QUESTION DIFFICULTY

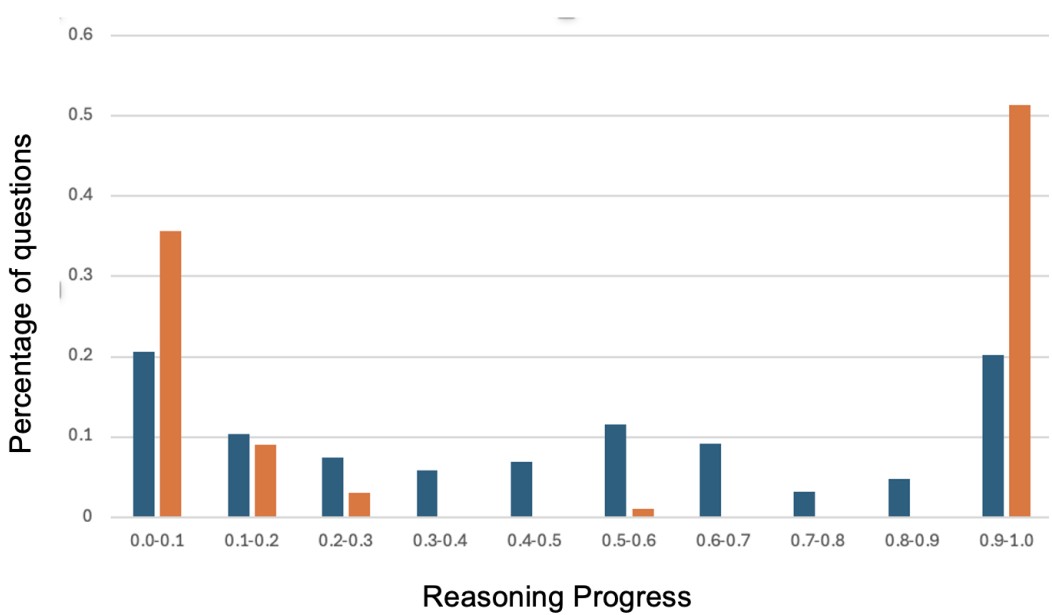

Figure Suppl. 1: Bin plot of reasoning progress on Math500. AdaptThink is in yellow and ESTAR-LITE in blue.

To better understand how models allocate reasoning effort, we constructed a bin plot where the x-axis represents reasoning progress (the fraction of reasoning steps relative to a full CoT), as derived from prompting off-the-shelf Qwen3-8B. As shown in Figure Suppl. 1, AdaptThink follows a bimodal distribution, with most questions concentrated at the beginning (x=0–0.1) or end (x=0.9–1.0) of the reasoning spectrum. In contrast, ESTAR-LITE demonstrates a more uniform distribution across the entire range. This suggests that ESTAR-LITE engages in reasoning more flexibly, adjusting its depth based on the demands of each question, while Adapthink tends to either reason minimally or commit to full chains without intermediate granularity.

We further tested whether reasoning length correlates with question difficulty using the Math500 dataset, which provides expert-labeled difficulty levels. Kendall's tau correlation between difficulty and reasoning progress was 0.490 for AdaptThink and 0.554 for ESTAR-LITE, showing that both models adjust reasoning based on difficulty, but ESTAR-LITE does so more reliably. This stronger alignment supports the view that ESTAR-LITE allocates effort more selectively, reasoning deeply only when necessary, rather than applying a fixed strategy.

