# OpenReview forum: "Efficient Inference with Large Reasoning Models"
_ICLR.cc/2026/Conference — Submitted to ICLR 2026_

### Official Review · Reviewer_eejB · 2025-10-31

**Soundness:** 2
**Presentation:** 2
**Contribution:** 2
**Rating:** 2
**Confidence:** 4

**Summary:**

This paper proposes ESTAR (Early-Stopping for Token-Aware Reasoning) to address the reasoning redundancy problem in Large Reasoning Models (LRMs). The main idea is to detect and reduce this redundancy using a LightGBM classifier. The tuning process includes SFT on a dataset with the <stop> token inserted, and the finetuned model is further trained using GRPO with a compute-aware reward function. ESTAR can largely reduce output tokens and maintain accuracy, providing a better accuracy-efficiency trade-off than other methods.

**Strengths:**

- This paper tackles a practical problem. Redundant reasoning is a major source of inefficiency in LRMs, and the goal of stopping early without sacrificing accuracy is a valuable insight.
- The multi-stage design of ESTAR is a strong point. Combining a lightweight classifier with a model trained to propose its own stop tokens is a clever way.
- The reported results are good. A 3.7x token reduction while maintaining 98.9% of the original accuracy would be a strong contribution.

**Weaknesses:**

- This paper mainly claims efficiency, but the experiments simply translate this to output tokens, which ignores the newly involved overhead. When using highly optimized inference engines like vllm or sglang, this method may break the workload, so I would like to see if there is end-to-end testing.
- The consistency and accuracy definitions may have problems and even bring contradiction in training. The LITE classifier and SFT focus on consistency, but the RL step is using an accuracy signal, which may conflict when the model's answer is wrong.
- The robustness of both training and evaluation is a huge concern. You mentioned temperature and top-p, but it seems only sampled once for both training and evaluation. Sampling can have a huge impact on reasoning output and even the final answer.
- As your method evolves through its 3 stages, there is a lack of ablations on how each component helps with the final result.
- The paper lacks details, especially for baseline methods. The repository you provided is also empty (as of 10/31/2025).

**Questions:**

- How does this method deal with batch scenarios?
- How do you deal with wrong answers, specifically in the LITE classifier step and the SFT step?
- Have you tried random sampling for the training data? How will different samples from the same question influence the result?
- Will the classifier be updated in the SFT and RL steps?

---

> ### Author Response · Authors · 2025-11-22
> **Response to Reviewer eejB**
>
> > [This paper mainly claims efficiency, but the experiments simply translate this to output tokens, which ignores the newly involved overhead. When using highly optimized inference engines like vllm or sglang, this method may break the workload, so I would like to see if there is end-to-end testing.]
>
> **Response:**
> We implement our algorithm on top of **vLLM** and modify the async engine to support our early-stop controller. For each request, the engine maintains a main *think stream* and an asynchronous *probe stream*. The main stream continuously decodes the chain-of-thought tokens, while the probe stream is periodically forked from the same internal state using vLLM's KV-cache sharing: the probe reuses the parent's prefix key–value blocks and only appends a short suffix that elicits a forced `<final_answer>` completion. The probe returns both the candidate answer and its token-level log-probabilities, which are fed into the lightweight classifier, while the main stream keeps generating `<think>` tokens in parallel. Once the classifier detects that a stopping condition is met, it sends a cancel signal to the main stream, after which we return the accumulated `<think>` trace together with the probe's final answer. Throughout this process the KV cache is shared and updated in a block-wise manner, so only the blocks corresponding to newly generated `<think>` tokens are written, keeping additional memory overhead minimal and preserving vLLM's batching and throughput advantages.
>
> | latency (seconds)      | Qwen3-4B | Qwen3-8B | Qwen3-14B | Qwen3-32B | QwQ-32B | DeepSeek-R1-Distill-Llama-8B | DeepSeek-R1-Distill-Qwen-7B |
> |------------------------|----------|----------|-----------|-----------|---------|-------------------------------|-----------------------------|
> | vanilla                | 8926     | 10658    | 11697     | 11795     | 10375   | 11245                         | 9238                        |
> | ours                   | 1911     | 2508     | 3131      | 4129      | 3963    | 2852                          | 2485                        |
> | speedup (vanilla/ours) | 4.67     | 4.25     | 3.74      | 2.86      | 2.62    | 3.94                          | 3.72                        |

---

> ### Author Response · Authors · 2025-11-22
> **Response to Reviewer eejB**
>
> > [The consistency and accuracy definitions may have problems and even bring contradiction in training. The LITE classifier and SFT focus on consistency, but the RL step is using an accuracy signal, which may conflict when the model's answer is wrong.]
>
> **Response:**
> We apologize for the confusion and will clarify the roles of the three stages. Conceptually, all components ultimately aim at *accurate* final answers; “consistency’’ is only used as a proxy for deciding *when it is safe to stop*.
>
> 1. **Classifier and SFT: consistency w.r.t. the model’s own final answer.**
>    The LITE classifier and the SFT stage operate on a *fixed* or only lightly tuned backbone LRM. In this setting, they cannot change whether the underlying model is correct or not; instead, their goal is to detect when the model has already converged to its *own* final answer so that we can avoid generating unnecessary tokens. Concretely, we define a consistency label at step $t$ if the forced answer after `</think>` at step $t$ equals the answer the same model would produce after a long CoT. The LITE classifier learns to predict this “already-converged’’ state, and SFT teaches the model to emit a special `<stop>` token at those positions. Both are therefore consistency-oriented by design and do not introduce any extra tension with task accuracy—they simply shorten trajectories of the current model.
>
> 2. **RL: accuracy-aware early stopping.**
>    In contrast, the GRPO-based RL stage is allowed to change the LRM itself and therefore legitimately optimizes for *accuracy* against the gold answer. Our reward couples these two objectives: a trajectory only receives a high reward if (i) the final forced answer is correct, and (ii) the `<stop>` token appears close to the earliest position where the answer has already stabilized. Thus, early stopping is never rewarded in isolation; an “early but wrong’’ answer is penalized by the accuracy term, so there is no incentive for the model to trade off correctness for shorter CoT. Instead, RL improves GRPO in two ways at once: it pushes the model toward producing the correct final answer, and it learns to place the `<stop>` token as soon as that correct answer is reachable.
>
> 3. **Why we do not enforce per-prefix accuracy supervision (efficiency).**
>    A naive way to define “consistency’’ would be to compare *every* candidate prefix’s answer to the eventual gold answer. However, doing so online is prohibitively expensive: to know the “final’’ answer for a candidate, one must first generate its full trajectory, and then, for each potential `<stop>` position, reason over the corresponding prefix. The effective cost per candidate would scale with the sum of the full length and all prefix lengths, which is quadratic in the CoT length. Our design avoids this inefficiency. We generate a long CoT once, extract the earliest converged position offline, and then, at training time, share this supervision across the classifier, SFT, and RL. At inference time, combined with our optimized asynchronous vLLM implementation, we only need a *single* pass: the main stream grows `<think>` while a lightweight probe produces the candidate answer; once the learned controller fires, we stop decoding immediately.
>
> In summary, consistency-based objectives (classifier + SFT) and accuracy-based RL do not contradict each other; they act at different levels. The former learns *when* the current model has already settled on its final answer, while the latter improves *what* that answer is and jointly encourages the model to emit a `<stop>` token as soon as the correct answer is available, yielding both higher accuracy and shorter CoT.

---

> ### Author Response · Authors · 2025-11-22
> **Response to Reviewer eejB**
>
> > [As your method evolves through its 3 stages, there is a lack of ablations on how each component helps with the final result.]
>
> **Response:**
> We appreciate the request for a clearer decomposition of how each stage contributes to the final performance. Our method contains 3 stages: In the first stage, we trained a lightweight classifier to determine early stopping (ESTAR-LITE). In the second stage, we further labeled some data and supervised finetuned the LLM to propose when to stop (ESTAR-FT). In the third stage, we further applied reinforcement learning with length penalty (ESTAR). Figure 4 already provides exactly this ablation study on MATH-500. It shows a clear, monotonic improvement from ESTAR-LITE to ESTAR-FT, then to ESTAR, moving toward the upper-left corner (earlier stops) with higher consistency.
>
> In the revised version, we will reorganize Section 6 to explicitly present these as three ablation levels. The experiment results will be added to table
>
>
> | Method         | USMLE Acc | USMLE Len | JAMA Acc | JAMA Len | MATH500 Acc | MATH500 Len | AIME Acc | AIME Len |
> |---------------|-----------|-----------|----------|----------|-------------|-------------|----------|----------|
> | GRPO          | 78.14     | 2732      | 57.8     | 2634     | 94.0        | 3962        | 70.00    | 9871     |
> |               | (100.0%)  | (x1.0)    | (100.0%) | (x1.0)   | (100.0%)    | (x1.0)      | (100.0%) | (x1.0)   |
> | No-Thinking   | 66.2      | 315       | 48.2     | 369      | 85.0        | 1139        | 26.67    | 1513     |
> |               | (84.7%)   | (x8.7)    | (83.4%)  | (x7.1)   | (90.4%)     | (x3.5)      | (38.1%)  | (x6.5)   |
> | AdaptThink    | 76.4      | 987       | 55.8     | 1102     | 93.8        | 2130        | 66.67    | 4513     |
> |               | (97.8%)   | (x2.8)    | (96.5%)  | (x2.4)   | (99.8%)     | (x1.9)      | (95.2%)  | (x2.2)   |
> | Length-Penalty| 76.6      | 1325      | 55.4     | 1872     | 92.4        | 3190        | 66.67    | 7324     |
> |               | (98.0%)   | (x2.1)    | (95.8%)  | (x1.4)   | (98.3%)     | (x1.2)      | (95.2%)  | (x1.3)   |
> | ESTAR-LITE    | 76.83     | 549       | 56.6     | 419      | 93.2        | 2019        | 66.67    | 3045     |
> |               | (98.3%)   | (x5.0)    | (97.8%)  | (x6.3)   | (99.1%)     | (x2.0)      | (95.2%)  | (x3.2)   |
> | ESTAR-FT      | 77.10     | 645       | 57.2     | 689      | 93.4        | 2401        | 66.67    | 3413     |
> |               | (98.7%)   | (x4.2)    | (99.0%)  | (x3.8)   | (99.4%)     | (x1.7)      | (95.2%)  | (x2.9)   |
> | ESTAR         | 77.13     | 388       | 56.10    | 352      | 93.8        | 635         | 70.00    | 3788     |
> |               | (98.7%)   | (x7.0)    | (97.1%)  | (x7.5)   | (99.8%)     | (x6.2)      | (100.0%) | (x2.6)   |

---

> ### Author Response · Authors · 2025-11-22
> **Response to Reviewer eejB**
>
> > [The paper lacks details, especially for baseline methods. The repository you provided is also empty (as of 10/31/2025).]
>
> Sorry that the code repository has been empty. We’ve updated the GitHub link, and the new one now contains the full, detailed code:
>
> https://anonymous.4open.science/r/Stop-Think-F789/
>
> For lack baseline methods, we conduct additional experiments on MATH-500 using QwQ-32B, following the evaluation protocol in [O1-Pruner](https://arxiv.org/html/2501.12570v1).
>
> We compare our method with 6 representative efficient-reasoning baselines, covering most of the efficiency techniques related to early stop, including length-based penalties methods:
>
> [1] Luo, Haotian, et al. "O1-pruner: Length-harmonizing fine-tuning for o1-like reasoning pruning, 2025." arXiv preprint arxiv:2501.12570 (2025).
>
> [2] Jiang, Guochao, et al. "Flashthink: An early exit method for efficient reasoning." arXiv preprint arXiv:2505.13949 (2025).
>
> [3] Chen, Runjin, et al. "Seal: Steerable reasoning calibration of large language models for free." arXiv preprint arXiv:2504.07986 (2025).
>
> [4] Yang, Chenxu, et al. "Dynamic early exit in reasoning models." arXiv preprint arXiv:2504.15895 (2025).
>
> [5] Zhang, Jiajie, et al. "Adaptthink: Reasoning models can learn when to think, 2025."
>
> [6] Liu, Xin, and Lu Wang. "Answer Convergence as a Signal for Early Stopping in Reasoning." arXiv preprint arXiv:2506.02536 (2025).
>
> We evaluate using 3 metrics:
> (1) accuracy, (2) CoT length reduction, and (3) Accuracy–Efficiency Score (AES), which evaluate the trade-off between improving accuracy and reducing CoT length. CoT length reduction is calculated as the difference between the length count of the vanilla and the proposed, divided by the vanilla length count. AES is defined as linearly combining the relative change in CoT length and the relative change in accuracy (we used the default value for hyperparameters \(\alpha\) and \(\beta\) from O1-Pruner):
>
> $$
> AES =
> \alpha \left( \frac{Length_{vanilla} - Length_{proposed}}{Length_{vanilla}} \right)
> +
> \beta \left( \frac{Acc_{proposed} - Acc_{vanilla}}{Acc_{vanilla}} \right)
> $$
>
> The result is shown below with top two performing methods highlighted. Our method achieves the highest AES while simultaneously improving accuracy and achieving one of the largest CoT length reductions.
>
> | Method           | Acc (Vanilla) | Acc (Proposed) | Length Reduction | AES |
> |------------------|------------------|-------------------|-----------------|---------------|
> | O1-Pruner        | 94.2             | 91.6              | 27.9\%          | 0.141         |
> | FlashThink       | 95.2             | **95.9**          | 23.0\%          | 0.252         |
> | DEER             | 93.8             | 94.6              | 26.4\%          | **0.290**     |
> | DEER-Pro         | 93.8             | 94.8              | 19.0\%          | 0.222         |
> | SEAL             | 94.4             | 95.0              | 18.8\%          | 0.207         |
> | LEARN TO STOP    | 95.0             | 93.0              | **31.6\%**      | 0.210         |
> | Adapthink        | 95.2             | **95.9**          | 25.2\%          | 0.274         |
> | Ours             | 95.8             | **96.2**          | **30.5\%**      | **0.318**     |

---

> > ### Author Response · Authors · 2025-11-22
> > **Response to Reviewer eejB**
> >
> > > [The robustness of both training and evaluation is a huge concern. You mentioned temperature and top-p, but it seems only sampled once for both training and evaluation. Sampling can have a huge impact on reasoning output and even the final answer.]
> >
> > **Response:**
> >
> > Each question is evaluated with four independent runs (different random seeds, identical decoding settings). To illustrate the run-to-run variance, we report the full per-run results for Qwen3-8B on MATH-500 below:
> >
> > | Method  | Run  | Accuracy | CoT Length (tokens) | Latency (seconds) |
> > |---------|------|----------|---------------------|-------------------|
> > | vanilla | run1 | 0.946    | 3981.1              | 10848.3           |
> > | vanilla | run2 | 0.936    | 3993.4              | 10996.3           |
> > | vanilla | run3 | 0.938    | 3911.8              | 10416.3           |
> > | vanilla | run4 | 0.938    | 3916.5              | 10372.5           |
> > | ours    | run1 | 0.932    | 1889.2              | 2429.9            |
> > | ours    | run2 | 0.936    | 1990.2              | 2447.4            |
> > | ours    | run3 | 0.936    | 2084.6              | 2523.1            |
> > | ours    | run4 | 0.928    | 2115.7              | 2632.8            |
> >
> > As shown, accuracy is highly stable across runs in both the vanilla and ESTAR settings (within ±0.4 percentage points), while ESTAR consistently reduces CoT length by about 2× and cuts latency by roughly 4×, with negligible impact on accuracy.
> >
> > > [How does this method deal with batch scenarios?]
> >
> > **Response:**
> > We implement our method on top of vLLM and extend its async engine to fully support batched inference. Within a batch, each example maintains its own main think stream and asynchronous probe stream, so sequences are decoded independently while still sharing vLLM’s batching efficiency. Both the main decoding and the probes are scheduled asynchronously, allowing the controller to trigger early stopping for some sequences in the batch while others continue generating, with probes also handled in an asynchronous, per-sample fashion.
> >
> >
> > > [How do you deal with wrong answers, specifically in the LITE classifier step and the SFT step?]
> >
> > **Response:**
> > For the LITE classifier and the SFT stage, our goal is not to correct the model’s mistakes but to decide when the model has already settled on its final answer. Both components are consistency-oriented rather than accuracy-improving.
> >
> > Concretely, if the underlying LRM produces a wrong answer on a question, this reflects the limitation of the base model itself. In that case, the role of LITE and SFT is simply to detect that the model has already converged to this (possibly incorrect) answer and to avoid wasting additional tokens on further, unproductive thinking. In other words, once the model’s answer is stable, we prefer to stop early rather than generate a longer chain-of-thought around an answer that is already wrong. Accuracy is handled by the base model (and the RL stage), while LITE and SFT focus purely on stopping as soon as the model’s own final answer is reachable.
> >
> > > [Will the classifier be updated in the SFT and RL steps?]
> >
> > **Response:**
> > In SFT and RL, our focus is on shaping the generator (the LRM) rather than the detector. In particular, during RL we already have access to the gold answer, so we can directly check whether a trajectory has converged to the correct answer and compute rewards accordingly, without needing the classifier at all. We won't use the classifier throughout SFT and RL, and only (re)train it after the backbone model has been fully updated.
> >
> > Empirically, we also experimented with jointly updating the classifier during training and found that it mainly increases training time without bringing noticeable gains over the simpler “train classifier at the end” strategy. This is why, in our final pipeline, the classifier is kept fixed in SFT/RL and trained separately once the model has converged.

---

### Official Review · Reviewer_qWyZ · 2025-10-31

**Soundness:** 3
**Presentation:** 1
**Contribution:** 3
**Rating:** 6
**Confidence:** 4

**Summary:**

The paper presents a novel three-stage approach called Early-Stopping for Token-Aware Reasoning (ESTAR) that detects and reduces reasoning redundancy without sacrificing accuracy for LRMS. The approach consists of: (1) Using a lightweight classifier (LightGBM) to identify optimal early-stopping positions in reasoning trajectories; (2) Training the LRM via Supervised Fine-Tuning (SFT) to autonomously generate `<stop>` signals based on these positions; and (3) Employing Reinforcement Learning (RL) to further refine the model's self-generation of these `<stop>` signals. Experimental results across four benchmarks indicate that ESTAR achieves a significant reduction in reasoning length (approximately 3.8x) while maintaining task accuracy.

**Strengths:**

1. The motivation is clear and the topic is well presented.

2. The proposed ESATR method is described in sufficient detail and appears technically sound.

3. The experimental results are strong, demonstrating the effectiveness of the proposed method.

**Weaknesses:**

### Major Concerns:

1. Overly General Title: The paper's title is too broad. "Efficient inference with LRMs" can be achieved through many different approaches (e.g., architectural changes). The current title does not accurately reflect the paper's contribution.

2. Empty Code Repository: The provided anonymous repository for code is currently empty.

3. Presentation and Formatting Issues: The core methodology sections (Preliminaries and Methods) suffer from presentation issues. The text appears "messy" and contains inconsistent formatting. Specific issues include:

- Awkward or unusual line breaks in the Preliminaries section.
- The use of non-standard markers, such as "(RQ1)", within the body of the Methods section, which seem out of place in a formal paper.
- Inconsistent formatting of mathematical equations between the Preliminaries and Methods sections.

4. Unclear Statements: Several statements in the paper are confusing or incomplete and require clarification:
- L123, ”... update ESTAR-LITE to stay aligned with the new trajectories.“
- L195, "... the tabular classifier also confirms the consistency of the model’s earlystop answer."
- L251, "Then we apply regular teacher ..."

### Minor Issues:

1. L121, ESTAR-FT is used before its formal definition.

2. L203, inconsistent notations vs. Eq. (1).

**Questions:**

The core methodology and the reported results are promising, and this work appears to be a valuable contribution to the community. However, the paper's current writing is a significant concern. In order to raise my rating, I would like the authors to address the different points in the major concerns.

---

> ### Author Response · Authors · 2025-11-22
> **Response to Reviewer qWyZ**
>
> > [Empty Code Repository: The provided anonymous repository for code is currently empty.]
>
> **Response:**
>
> Sorry that the code repository has been empty. We’ve updated the GitHub link, and the new one now contains the full, detailed code:
>
> https://anonymous.4open.science/r/Stop-Think-F789/
>
> We implement our algorithm on top of **vLLM** and modify the async engine to support our early-stop controller. For each request, the engine maintains a main *think stream* and an asynchronous *probe stream*. The main stream continuously decodes the chain-of-thought tokens, while the probe stream is periodically forked from the same internal state using vLLM's KV-cache sharing: the probe reuses the parent's prefix key–value blocks and only appends a short suffix that elicits a forced `<final_answer>` completion. The probe returns both the candidate answer and its token-level log-probabilities, which are fed into the lightweight classifier, while the main stream keeps generating `<think>` tokens in parallel. Once the classifier detects that a stopping condition is met, it sends a cancel signal to the main stream, after which we return the accumulated `<think>` trace together with the probe's final answer. Throughout this process the KV cache is shared and updated in a block-wise manner, so only the blocks corresponding to newly generated `<think>` tokens are written, keeping additional memory overhead minimal and preserving vLLM's batching and throughput advantages.
>
>
> | latency (seconds)      | Qwen3-4B | Qwen3-8B | Qwen3-14B | Qwen3-32B | QwQ-32B | DeepSeek-R1-Distill-Llama-8B | DeepSeek-R1-Distill-Qwen-7B |
> |------------------------|----------|----------|-----------|-----------|---------|-------------------------------|-----------------------------|
> | vanilla                | 8926     | 10658    | 11697     | 11795     | 10375   | 11245                         | 9238                        |
> | ours                   | 1911     | 2508     | 3131      | 4129      | 3963    | 2852                          | 2485                        |
> | speedup (vanilla/ours) | 4.67     | 4.25     | 3.74      | 2.86      | 2.62    | 3.94                          | 3.72                        |

---

> ### Author Response · Authors · 2025-11-22
> **Response to Reviewer qWyZ**
>
> > [Overly General Title: The paper's title is too broad. "Efficient inference with LRMs" can be achieved through many different approaches (e.g., architectural changes). The current title does not accurately reflect the paper's contribution.]
>
> **Response:**
> Thanks for pointing out the issue with the title. To aligned with our paper’s contributions, we will change it to - ESTAR: Classifier-Verified Early Stopping for Efficient Chain-of-Thought Reasoning.
>
> > [Presentation and Formatting Issues: The core methodology sections (Preliminaries and Methods) suffer from presentation issues. The text appears "messy" and contains inconsistent formatting.]
>
> **Response:**
>
> Thank you for pointing this out. We appreciate your suggestions regarding presentation and formatting. We will revise the Preliminaries and Methods sections to improve clarity and consistency, and we will upload an updated PDF with these changes within the next week.
>
>
>
>
> > [Unclear Statements:
> L123, ”... update ESTAR-LITE to stay aligned with the new trajectories.“]
>
> **Response:**
> By "update ESTAR-LITE" in this sentenc, we mean that after RL modifies the distribution of reasoning trajectories, we regenerate the early-stop supervision signals and retrain the LightGBM classifier on these updated trajectories. This ensures the classifier remains calibrated to the model’s current reasoning patterns, rather than the pre-RL distribution.
> Concretely, after each RL epoch, we:
>
> 1. collect the new trajectories generated by the RL policy;
>
> 2. recompute the features (logit slopes, cumulative evidence, stability metrics, etc.);
>
> 3. relabel safe/unsafe stop points using full rollout answers;
>
> 4. retrain the ESTAR-LITE LightGBM classifier once per epoch.
>
>
> > [Unclear Statements: L195, "... the tabular classifier also confirms the consistency of the model’s earlystop answer."]
>
> **Response:**
> Thanks for pointing out the unclearness of what exactly the classifier is "confirming".
>
> At inference time, the LRM first proposes a `<stop>` token. We then apply the classifier, which operates only on token-level feature signals, to predict whether stopping at this position would preserve the final answer the LRM would have produced with a full chain-of-thought. If the classifier predicts the stop is safe, we truncate the reasoning; otherwise, the LRM continues thinking.
>
>
> > [Unclear Statements: L251, "Then we apply regular teacher ..."]
>
> **Response:**
> After inserting `<stop>` tokens into the CoT, we perform standard next-token supervised finetuning over the entire formatted sequence. The model is trained with teacher forcing to predict every subsequent token, including all reasoning tokens, the inserted `<stop>` markers, `</think>`, and the final answer. The `<stop>` token is added to the model vocabulary and treated identically to other tokens during SFT. We do not mask or reweight any part of the sequence: the model simply learns, through next-token prediction, to output `<stop>` exactly at the annotated positions where forced early-stop evaluation confirmed that the intermediate answer is already consistent with the final one.

---

### Official Review · Reviewer_oRZx · 2025-11-01

**Soundness:** 2
**Presentation:** 2
**Contribution:** 2
**Rating:** 2
**Confidence:** 4

**Summary:**

This work combines truncation-base and fine-tuning-base approaches for efficient reasoning.

**Strengths:**

- Efficient reasoning is one of the more important fields for LRMs.
- Testing consistency + accuracy is a good point. I am convinced.

**Weaknesses:**

The experiment execution seems to be on the weaker side.
- Only evaluated on two Qwen3 models.
- Baselines are mainly only featured in Table 2, but not all tasks and models.
- Questionable decoding budget as indicated in Table 1—the vanilla decoding is often <5k.
- No end-to-end latency/throughput efficiency report.

Also, there should be more discussion and comparison of other efficient reasoning methods. Without digging too much, this work is already missing some key comparisons.
- o1-pruner is a well-recognized fine-tuning-based method for efficient reasoning.
- FlashThink, which is cited in the paper, is in fact not a binary (think or no think) method but a truncation one. Similarly, https://arxiv.org/abs/2506.02536 is another one. There are definitely many more early stopping methods applied to CoT, and they should be properly discussed and compared.
- AutoL2S also utilized the idea of incorporating special token for efficient reasoning, which should also be compared and discussed.

**Questions:**

- Which model is Table 2 testing?
- How many run per each question?

---

> ### Author Response · Authors · 2025-11-22
> **Response to Reviewer oRZx**
>
> > [Only evaluated on two Qwen3 models.]
>
> **Response:**
> Qwen3 reasoning models exhibit long-CoT behaviors similar to DeepSeek/Qwen-QwQ. We selected them because they are open, reproducible, and representative of LRMs with thinking mode. ESTAR does not rely on any Qwen-specific architecture or training signal.
>
> Nonetheless, we added a new experiment on 5 additional open LRMs in the revision to further validate model-agnosticity.
> Using VLLM, we evaluated each model on Math-500 under both the vanilla settings and our early stop method.
> We report both Token Reduction\% and Accuracy\% as the evaluation metrics. Token Reduction\% is calculated as the difference between the token count of the vanilla setting and our method, divided by the vanilla token count. Accuracy improvement\% is defined as the accuracy of our method divided by that of the vanilla model.
> Results are consistent: we reduced CoT length usage by about 37\% compared to the vanilla model, while maintaining about the same accuracy.
>
> | Model                          | Tokens (Vanilla) | Tokens (Ours) | Token Reduction \%   | Accuracy (Vanilla) | Accuracy (Ours) | Accuracy improvement\% |
> |--------------------------------|-----------------|--------------|----------|--------------------|-----------------|------------|
> | Qwen3-4b                       | 3193            | 1771         | 44.5\%     | 0.924              | 0.916           | 99.1\%      |
> | Qwen3-32b                      | 3285            | 2076         | 36.8\%     | 0.974              | 0.970           | 99.6\%      |
> | QwQ-32b                        | 3848            | 2673         | 30.5\%     | 0.958              | 0.962           | 100.4\%     |
> | DeepSeek-R1-Distill-Llama-8B   | 4173            | 2909         | 30.3\%     | 0.890              | 0.890           | 100.0\%     |
> | DeepSeek-R1-Distill-Qwen-7B    | 2994            | 1716         | 42.7\%     | 0.928              | 0.930           | 100.2\%     |

---

> > ### Author Response · Authors · 2025-11-22
> > **Response to Reviewer oRZx**
> >
> > > [Baselines are mainly only featured in Table 2 but not all tasks.]
> >
> > **Response:**
> > Thanks for pointing out that Table 2 missed GPQA. However, the comparison on GPQA in Table 2 is not feasible because Table 1 and Table 2 evaluate different settings. In Table 1 (ESTAR-LITE), only the lightweight classifier is trained—on USMLE-QA—while the underlying LRM remains unchanged. This allows evaluation on GPQA as a true out-of-domain test of classifier generalization. In contrast, Table 2 reports results for the full ESTAR system after training the LRM itself on domain-specific data (USMLE-QA for Closed QA or DEEPSCALER for Open QA). Once the LRM is fine-tuned on medical domain, its accuracy drops on the STEM-heavy GPQA benchmark (e.g., ESTAR 50.2\% vs. Qwen3-8B 60.10\%; AdaptThink similarly drops to 46.0\%). Including these GPQA numbers in Table 2 would therefore conflate domain shift with the effects of efficient-reasoning methods and result in misleading comparisons.
> >
> > For clarity, we will explicitly state this experimental design choice in the revision. We will add a short explanation in the main text clarifying that: (1) Table 1 evaluates ESTAR-LITE without modifying the underlying LRM and uses GPQA to assess cross-domain generalization; and (2) Table 2 evaluates the full ESTAR pipeline after LRM post-training, which is domain-specific, and therefore excludes out-of-domain datasets like GPQA to avoid misinterpretation.

---

> > > ### Author Response · Authors · 2025-11-22
> > > **Response to Reviewer oRZx**
> > >
> > > > [Questionable decoding budget as indicated in Table 1—the vanilla decoding is often <5k.]
> > >
> > > **Response:**
> > > We would like to clarify that the vanilla chain-of-thought lengths reported in Table 1 are not artificially inflated, nor do we apply any decoding tricks to lengthen the baseline. We follow the official Qwen3 recommended reasoning mode sampling settings: temperature = 0.6, top-k = 20, top-p = 0.95, and repetition penalty 1.2. These settings are identical to the [public Qwen3 inference examples](https://huggingface.co/Qwen/Qwen3-8B).
> > >
> > > Importantly, our baseline CoT lengths are fully consistent with prior work. For example, in [AutoL2S](https://arxiv.org/pdf/2501.12570) reviewer referenced, the reported Qwen2.5-7B-Instruct baseline reasoning length is 4,483 tokens, [DEER](https://arxiv.org/pdf/2504.15895v3) also reported a QwQ-32B baseline of 4,508 tokens, while our Qwen3-8B baseline length on MATH500 is 3,963 tokens, which are all under 5k tokens. This shows that the decoding budget we used is typical for open-source LRMs when thinking-mode is enabled and is well within the expected range (<5k).

---

> ### Author Response · Authors · 2025-11-22
> **Response to Reviewer oRZx**
>
> > [No end-to-end latency/throughput efficiency report.]
>
> We implement our algorithm on top of **vLLM** and modify the async engine to support our early-stop controller. For each request, the engine maintains a main *think stream* and an asynchronous *probe stream*. The main stream continuously decodes the chain-of-thought tokens, while the probe stream is periodically forked from the same internal state using vLLM's KV-cache sharing: the probe reuses the parent's prefix key–value blocks and only appends a short suffix that elicits a forced `<final_answer>` completion. The probe returns both the candidate answer and its token-level log-probabilities, which are fed into the lightweight classifier, while the main stream keeps generating `<think>` tokens in parallel. Once the classifier detects that a stopping condition is met, it sends a cancel signal to the main stream, after which we return the accumulated `<think>` trace together with the probe's final answer. Throughout this process the KV cache is shared and updated in a block-wise manner, so only the blocks corresponding to newly generated `<think>` tokens are written, keeping additional memory overhead minimal and preserving vLLM's batching and throughput advantages.
>
> | latency (seconds)      | Qwen3-4B | Qwen3-8B | Qwen3-14B | Qwen3-32B | QwQ-32B | DeepSeek-R1-Distill-Llama-8B | DeepSeek-R1-Distill-Qwen-7B |
> |------------------------|----------|----------|-----------|-----------|---------|-------------------------------|-----------------------------|
> | vanilla                | 8926     | 10658    | 11697     | 11795     | 10375   | 11245                         | 9238                        |
> | ours                   | 1911     | 2508     | 3131      | 4129      | 3963    | 2852                          | 2485                        |
> | speedup (vanilla/ours) | x4.67     | x4.25     | x3.74      | x2.86      | x2.62    | x3.94                          | x3.72                        |

---

> > ### Author Response · Authors · 2025-11-22
> > **Response to Reviewer oRZx**
> >
> > > [Lack of comparison to other efficient reasoning methods]
> >
> > To address this, we conduct additional experiments on MATH-500 using QwQ-32B, following the evaluation protocol in [O1-Pruner](https://arxiv.org/html/2501.12570v1).
> >
> > We compare our method with 6 representative efficient-reasoning baselines, covering most of the efficiency techniques related to early stop, including length-based penalties methods:
> > [1] Luo, Haotian, et al. "O1-pruner: Length-harmonizing fine-tuning for o1-like reasoning pruning, 2025." URL https://arxiv. org/abs/2501.12570 (2025).
> > [2] Jiang, Guochao, et al. "Flashthink: An early exit method for efficient reasoning." arXiv preprint arXiv:2505.13949 (2025).
> > [3] Chen, Runjin, et al. "Seal: Steerable reasoning calibration of large language models for free." arXiv preprint arXiv:2504.07986 (2025).
> > [4] Yang, Chenxu, et al. "Dynamic early exit in reasoning models." arXiv preprint arXiv:2504.15895 (2025).
> > [5] Zhang, Jiajie, et al. "Adaptthink: Reasoning models can learn when to think, 2025."
> > [6] Liu, Xin, and Lu Wang. "Answer Convergence as a Signal for Early Stopping in Reasoning." arXiv preprint arXiv:2506.02536 (2025).
> >
> > We evaluate using 3 metrics:
> > (1) accuracy, (2) CoT length reduction, and (3) Accuracy–Efficiency Score (AES), which evaluate the trade-off between improving accuracy and reducing CoT length. CoT length reduction is calculated as the difference between the length count of the vanilla and the proposed, divided by the vanilla length count. AES is defined as linearly combining the relative change in CoT length and the relative change in accuracy (we used the default value for hyperparameters \(\alpha\) and \(\beta\) from O1-Pruner):
> >
> > $$
> > AES =
> > \alpha \left( \frac{Length_{vanilla} - Length_{proposed}}{Length_{vanilla}} \right)
> > +
> > \beta \left( \frac{Acc_{proposed} - Acc_{vanilla}}{Acc_{vanilla}} \right)
> > $$
> >
> > The result is shown below with top two performing methods highlighted. Our method achieves the highest AES while simultaneously improving accuracy and achieving one of the largest CoT length reductions.
> >
> > | Method           | Acc (Vanilla) | Acc (Proposed) | Length Reduction | AES |
> > |------------------|------------------|-------------------|-----------------|---------------|
> > | O1-Pruner        | 94.2             | 91.6              | 27.9\%          | 0.141         |
> > | FlashThink       | 95.2             | **95.9**          | 23.0\%          | 0.252         |
> > | DEER             | 93.8             | 94.6              | 26.4\%          | **0.290**     |
> > | DEER-Pro         | 93.8             | 94.8              | 19.0\%          | 0.222         |
> > | SEAL             | 94.4             | 95.0              | 18.8\%          | 0.207         |
> > | LEARN TO STOP    | 95.0             | 93.0              | **31.6\%**      | 0.210         |
> > | Adapthink        | 95.2             | **95.9**          | 25.2\%          | 0.274         |
> > | Ours             | 95.8             | **96.2**          | **30.5\%**      | **0.318**     |

---

> > > ### Author Response · Authors · 2025-11-22
> > > **Response to Reviewer oRZx**
> > >
> > > > [How many run per each question?]
> > >
> > > **Response:**
> > >
> > >
> > > Each question is evaluated with four independent runs (different random seeds, identical decoding settings). To illustrate the run-to-run variance, we report the full per-run results for Qwen3-8B on MATH-500 below:
> > >
> > > | Method  | Run  | Accuracy | CoT Length (tokens) | Latency (seconds) |
> > > |---------|------|----------|---------------------|-------------------|
> > > | vanilla | run1 | 0.946    | 3981.1              | 10848.3           |
> > > | vanilla | run2 | 0.936    | 3993.4              | 10996.3           |
> > > | vanilla | run3 | 0.938    | 3911.8              | 10416.3           |
> > > | vanilla | run4 | 0.938    | 3916.5              | 10372.5           |
> > > | ours    | run1 | 0.932    | 1889.2              | 2429.9            |
> > > | ours    | run2 | 0.936    | 1990.2              | 2447.4            |
> > > | ours    | run3 | 0.936    | 2084.6              | 2523.1            |
> > > | ours    | run4 | 0.928    | 2115.7              | 2632.8            |
> > >
> > > As shown, accuracy is highly stable across runs in both the vanilla and ESTAR settings (within ±0.4 percentage points), while ESTAR consistently reduces CoT length by about 2× and cuts latency by roughly 4×, with negligible impact on accuracy.

---

### Official Review · Reviewer_KAMF · 2025-11-05

**Soundness:** 2
**Presentation:** 3
**Contribution:** 2
**Rating:** 4
**Confidence:** 4

**Summary:**

This paper focuses on improving the reasoning efficiency of large reasoning models (LRMs). First, ESTAR-LITE trains a classifier to predict whether reasoning should stop or continue. Then ESTAR-FT fine-tunes LRMs on curated CoT to enable them to determine their own stopping points. Finally, ESTAR adapts GRPO to reward correct "stop" emissions, resulting in a more efficient reasoning model.

**Strengths:**

-	The paper is well-written and easy to follow.
-	The three research questions (when to truncate reasoning, how to let LRMs decide stopping points, and how to leverage self-generated stop signals with reinforcement learning) are well articulated and explored.
-	Experimental results are promising, demonstrating substantial reductions in tokens while maintaining original accuracy levels.

**Weaknesses:**

- Limited evaluations. Experiments are conducted only on Qwen3-8B and Qwen3-14B. Evaluations on additional LRMs would strengthen the conclusions. The results in Tables 1 and 2 appear to be repeated. Moreover, Table 2 seems to include only the results of Qwen3-8B. It would be helpful to also compare different methods with Qwen3-14B and other models (e.g., DeepSeek-R1).
- Limited discussion of baseline methods (AdaptThink and Length-Penalty). It would be helpful to clarify whether these are state-of-the-art approaches and to justify their selection as baselines. It would also be beneficial to compare with additional baseline methods, such as [1, 2, 3].
- Insufficient evaluation of reasoning quality. While efficiency has been improved, it would be useful to assess whether shortened reasoning remains coherent and logically sound.
- Minor: Figure 3 is somewhat difficult to read.

[1] Yang, C., Si, Q., Duan, Y., Zhu, Z., Zhu, C., Li, Q., ... & Wang, W. (2025). Dynamic Early Exit in Reasoning Models. arXiv preprint arXiv:2504.15895.

[2] Chen, R., Zhang, Z., Hong, J., Kundu, S., & Wang, Z. (2025). Seal: Steerable reasoning calibration of large language models for free. arXiv preprint arXiv:2504.07986.

[3] Wang, C., Feng, Y., Chen, D., Chu, Z., Krishna, R., & Zhou, T. (2025). Wait, We Don't Need to" Wait"! Removing Thinking Tokens Improves Reasoning Efficiency. arXiv preprint arXiv:2506.08343.

**Questions:**

-	In Table 2, why are comparisons across different methods on the GPQA dataset missing?

---

> ### Author Response · Authors · 2025-11-22
> **Response to Reviewer KAMF**
>
> > [Limited evaluations. Experiments are conducted only on Qwen3-8B and Qwen3-14B. Evaluations on additional LRMs would strengthen the conclusions. The results in Tables 1 and 2 appear to be repeated. Moreover, Table 2 seems to include only the results of Qwen3-8B. It would be helpful to also compare different methods with Qwen3-14B and other models (e.g., DeepSeek-R1).]
>
> **Response:**
> Qwen3 reasoning models exhibit long-CoT behaviors similar to DeepSeek/Qwen-QwQ. We selected them because they are open, reproducible, and representative of LRMs with thinking mode. ESTAR does not rely on any Qwen-specific architecture or training signal.
>
> Nonetheless, we added a new experiment on 5 additional open LRMs in the revision to further validate model-agnosticity.
> Using VLLM, we evaluated each model on Math-500 under both the vanilla settings and our early stop method.
> We report both Token Reduction\% and Accuracy\% as the evaluation metrics. Token Reduction\% is calculated as the difference between the token count of the vanilla setting and our method, divided by the vanilla token count. Accuracy improvement\% is defined as the accuracy of our method divided by that of the vanilla model.
> Results are consistent: we reduced CoT length usage by about 37\% compared to the vanilla model, while maintaining about the same accuracy.
>
> | Model                          | Tokens (Vanilla) | Tokens (Ours) | Token Reduction \%   | Accuracy (Vanilla) | Accuracy (Ours) | Accuracy improvement\% |
> |--------------------------------|-----------------|--------------|----------|--------------------|-----------------|------------|
> | Qwen3-4b                       | 3193            | 1771         | 44.5\%     | 0.924              | 0.916           | 99.1\%      |
> | Qwen3-32b                      | 3285            | 2076         | 36.8\%     | 0.974              | 0.970           | 99.6\%      |
> | QwQ-32b                        | 3848            | 2673         | 30.5\%     | 0.958              | 0.962           | 100.4\%     |
> | DeepSeek-R1-Distill-Llama-8B   | 4173            | 2909         | 30.3\%     | 0.890              | 0.890           | 100.0\%     |
> | DeepSeek-R1-Distill-Qwen-7B    | 2994            | 1716         | 42.7\%     | 0.928              | 0.930           | 100.2\%     |

---

> ### Author Response · Authors · 2025-11-22
> **Response to Reviewer KAMF**
>
> > [Limited discussion of baseline methods (AdaptThink and Length-Penalty). It would also be beneficial to compare with additional baseline methods, such as [1, 2, 3].]
>
> **Response:**
> Thank you for pointing out the need for a broader comparison with existing efficient reasoning techniques.
> To address this, we conduct additional experiments on MATH-500 using QwQ-32B, following the evaluation protocol in [O1-Pruner](https://arxiv.org/html/2501.12570v1).
>
> We compare our method with 6 representative efficient-reasoning baselines, covering most of the efficiency techniques related to early stop, including length-based penalties methods:
>
> [1] Luo, Haotian, et al. "O1-pruner: Length-harmonizing fine-tuning for o1-like reasoning pruning, 2025." arXiv preprint arxiv:2501.12570 (2025).
>
> [2] Jiang, Guochao, et al. "Flashthink: An early exit method for efficient reasoning." arXiv preprint arXiv:2505.13949 (2025).
>
> [3] Chen, Runjin, et al. "Seal: Steerable reasoning calibration of large language models for free." arXiv preprint arXiv:2504.07986 (2025).
>
> [4] Yang, Chenxu, et al. "Dynamic early exit in reasoning models." arXiv preprint arXiv:2504.15895 (2025).
>
> [5] Zhang, Jiajie, et al. "Adaptthink: Reasoning models can learn when to think, 2025."
>
> [6] Liu, Xin, and Lu Wang. "Answer Convergence as a Signal for Early Stopping in Reasoning." arXiv preprint arXiv:2506.02536 (2025).
>
> We evaluate using 3 metrics:
> (1) accuracy, (2) CoT length reduction, and (3) Accuracy–Efficiency Score (AES), which evaluate the trade-off between improving accuracy and reducing CoT length. CoT length reduction is calculated as the difference between the length count of the vanilla and the proposed, divided by the vanilla length count. AES is defined as linearly combining the relative change in CoT length and the relative change in accuracy (we used the default value for hyperparameters \(\alpha\) and \(\beta\) from O1-Pruner):
>
> $$
> AES =
> \alpha \left( \frac{Length_{vanilla} - Length_{proposed}}{Length_{vanilla}} \right)
> +
> \beta \left( \frac{Acc_{proposed} - Acc_{vanilla}}{Acc_{vanilla}} \right)
> $$
>
> The result is shown below with top two performing methods highlighted. Our method achieves the highest AES while simultaneously improving accuracy and achieving one of the largest CoT length reductions.
>
> | Method           | Acc (Vanilla) | Acc (Proposed) | Length Reduction | AES |
> |------------------|------------------|-------------------|-----------------|---------------|
> | O1-Pruner        | 94.2             | 91.6              | 27.9\%          | 0.141         |
> | FlashThink       | 95.2             | **95.9**          | 23.0\%          | 0.252         |
> | DEER             | 93.8             | 94.6              | 26.4\%          | **0.290**     |
> | DEER-Pro         | 93.8             | 94.8              | 19.0\%          | 0.222         |
> | SEAL             | 94.4             | 95.0              | 18.8\%          | 0.207         |
> | LEARN TO STOP    | 95.0             | 93.0              | **31.6\%**      | 0.210         |
> | Adapthink        | 95.2             | **95.9**          | 25.2\%          | 0.274         |
> | Ours             | 95.8             | **96.2**          | **30.5\%**      | **0.318**     |

---

> > ### Author Response · Authors · 2025-11-22
> > **Response to Reviewer KAMF**
> >
> > > [Insufficient evaluation of reasoning quality. While efficiency has been improved, it would be useful to assess whether shortened reasoning remains coherent and logically sound.]
> >
> > **Response:**
> >
> > To further address the reviewer’s concern regarding the evaluation of reasoning quality, we added an assessment of logical correctness of the generated reasoning on the USMLE dataset. Our approach is inspired by recent prior work showing that analyzing the internal logical structure of model-generated explanations is an effective way to evaluate reasoning reliability [1,2].
> >
> > For each question, we collected reasoning traces from both (1) the vanilla full chain-of-thought and (2) our efficient reasoning method (ESTAR). We then used GPT-5 as an external evaluator, following evidence that stronger evaluators improve assessment reliability [3]. The evaluator was asked to determine whether the provided reasoning remained logically correct, including whether the steps were coherent, aligned with the problem context, and free of major logical flaws. This general notion of logical soundness is consistent with the evaluation frameworks discussed in prior studies of reasoning quality [1,2].
> >
> > To ensure the robustness of the evaluation, we instructed GPT-5 to provide rationale for the assessment, and two human annotators looked at these rationales and correct assessments if wrong, with >95\% agreement; only 9 (<1\%) judgments required correction. Following previous work [2], we reported the Logical Correct Rate (LCR), defined as the percentage of cases where the judge considers as logically correct.
> >
> > | Method           | LCR (in \%)       |
> > |------------------|-------------------|
> > | Vanilla full CoT | 84.6              |
> > | Our early stop   | 88.5              |
> >
> > The LCR increases from 84.6\% (vanilla) to 88.5\% (ours), indicating that our shorter reasoning is more logically consistent. This aligns with previous fidning that truncating redundant or self-doubting segments can reduce contradiction frequency[2].
> >
> > References:
> >
> > [1] Liu, Ziyi, et al. "Self-contradictory reasoning evaluation and detection." Findings of the Association for Computational Linguistics: EMNLP 2024. 2024.
> >
> > [2] Mündler, Niels, et al. "Self-contradictory hallucinations of large language models: Evaluation, detection and mitigation." arXiv preprint arXiv:2305.15852 (2023).
> >
> > [3] Chen, Nuo, et al. "Judgelrm: Large reasoning models as a judge." arXiv preprint arXiv:2504.00050 (2025).

---

> ### Author Response · Authors · 2025-11-22
> **Response to Reviewer KAMF**
>
> > [Minor: Figure 3 is somewhat difficult to read.]
>
> **Response:**
> Figure 3 illustrates a separation between CoT steps whose answer already matches the final answer and CoT steps that do not, since the distributions of `slope recent` and `delta recent` for match versus mismatch steps cluster in clearly different regions with minimal overlap, and their means are visibly far apart. This qualitative separation is quantitatively confirmed by the high AUROC and large Cohen’s d reported in each panel, showing that a randomly chosen match step almost always receives a higher feature value than a mismatch step. Together, these visual and numerical signals demonstrate that both features reliably distinguish converged reasoning steps from unstable ones.
>
> We thank the reviewer for pointing out that Figure 3 is difficult to read. We agree that the current visualization may cause readability issues for several reasons:
>
> 1. The panels use overlapping density-normalized histograms for the two classes (match/mismatch), which can produce visually cluttered overlaps and make differences harder to see.
>
> 2. The font of x axis and y axis is small, especially when the figure is scaled down.
>
> 3. The dashed versus dotted vertical mean lines are subtle and difficult to distinguish.
>
> To address this, we will revise the figure by increasing contrast between classes, enlarging fonts, and simplifying layout so that the separation between the two classes is visually clearer.
>
>
>
>
> > [In Table 2, why are comparisons across different methods on the GPQA dataset missing?]
>
> **Response:**
> The comparison on GPQA in Table 2 is not feasible because Table 1 and Table 2 evaluate different settings. In Table 1 (ESTAR-LITE), only the lightweight classifier is trained—on USMLE-QA—while the underlying LRM remains unchanged. This allows evaluation on GPQA as a true out-of-domain test of classifier generalization. In contrast, Table 2 reports results for the full ESTAR system after training the LRM itself on domain-specific data (USMLE-QA for Closed QA or DEEPSCALER for Open QA). Once the LRM is fine-tuned on medical domain, its accuracy drops on the STEM-heavy GPQA benchmark (e.g., ESTAR 50.2\% vs. Qwen3-8B 60.10\%; AdaptThink similarly drops to 46.0\%). Including these GPQA numbers in Table 2 would therefore conflate domain shift with the effects of efficient-reasoning methods and result in misleading comparisons.
>
> For clarity, we will explicitly state this experimental design choice in the revision. We will add a short explanation in the main text clarifying that: (1) Table 1 evaluates ESTAR-LITE without modifying the underlying LRM and uses GPQA to assess cross-domain generalization; and (2) Table 2 evaluates the full ESTAR pipeline after LRM post-training, which is domain-specific, and therefore excludes out-of-domain datasets like GPQA to avoid misinterpretation.

---

### Meta-Review · Area_Chair_gaKL · 2025-12-12

**Summary:**

One major concern raised by the reviewers is that while the title is big, "efficient inference with Large Reasoning Models", the empirical studies are weak: 1. The sizes of the models used for evaluation are relatively small. 2. Some related baselines are missing for comparison. 3. Some detailed experimental setups are not clear.

**Reviewer Concerns:**

The authors' revisions remain limited and unconvincing:
1. Experiments with larger models and additional baselines were each conducted on only a single dataset, limiting the generalizability of the findings.
2. By comparing Acc (vanilla) and Acc (proposed) between the authors' method and the baselines in the additional experiments, the claim of superior performance of the proposed method is not well supported.
3. The demonstration of variance across three runs, though positive, is also restricted to a single dataset.

**Reviewer Scores:**

As most of the critical concerns remain, I do think the reviewers would change their scores.

---

### Decision · Program_Chairs · 2026-01-26

Reject